# A U-turn on Double Descent: Rethinking Parameter Counting in Statistical Learning

**Alicia Curth**[*]
University of Cambridge
amc253@cam.ac.uk

**Alan Jeffares**[*]
University of Cambridge
aj659@cam.ac.uk

**Mihaela van der Schaar**
University of Cambridge
mv472@cam.ac.uk

## Abstract

Conventional statistical wisdom established a well-understood relationship between model complexity and prediction error, typically presented as a *U-shaped curve* reflecting a transition between under- and overfitting regimes. However, motivated by the success of overparametrized neural networks, recent influential work has suggested this theory to be generally incomplete, introducing an additional regime that exhibits a second descent in test error as the parameter count $p$ grows past sample size $n$ – a phenomenon dubbed *double descent*. While most attention has naturally been given to the deep-learning setting, double descent was shown to emerge more generally across non-neural models: known cases include *linear regression, trees, and boosting*. In this work, we take a closer look at the evidence surrounding these more classical statistical machine learning methods and challenge the claim that observed cases of double descent truly extend the limits of a traditional U-shaped complexity-generalization curve therein. We show that once careful consideration is given to *what is being plotted* on the x-axes of their double descent plots, it becomes apparent that there are implicitly multiple, distinct complexity axes along which the parameter count grows. We demonstrate that the second descent appears exactly (and *only*) when and where the transition between these underlying axes occurs, and that its location is thus *not* inherently tied to the interpolation threshold $p = n$. We then gain further insight by adopting a classical nonparametric statistics perspective. We interpret the investigated methods as *smoothers* and propose a generalized measure for the *effective* number of parameters they use *on unseen examples*, using which we find that their apparent double descent curves do indeed fold back into more traditional convex shapes – providing a resolution to the ostensible tension between double descent and traditional statistical intuition.

## 1 Introduction

Historically, throughout the statistical learning literature, the relationship between model complexity and prediction error has been well-understood as a careful balancing act between *underfitting*, associated with models of high bias, and *overfitting*, associated with high model variability. This implied tradeoff, with optimal performance achieved between extremes, gives rise to a U-shaped curve, illustrated in the left panel of Fig. 1. It has been a fundamental tenet of learning from data, omnipresent in introductions to statistical learning [HT90, Vap95, HTF09], and is also practically reflected in numerous classical model selection criteria that explicitly trade off training error with model complexity [Mal73, Aka74, Sch78]. Importantly, much of the intuition relating to this U-shaped curve was originally developed in the context of the earlier statistics literature (see e.g. the historical note in [Nea19]), which focussed on conceptually simple learning methods such as linear regression, splines or nearest neighbor methods [WW75, HT90, GBD92] and their expected *in-sample* prediction error, which fixes inputs and resamples noisy outcomes [Mal73, HT90, RT19].

---

[*]Equal contribution

37th Conference on Neural Information Processing Systems (NeurIPS 2023).

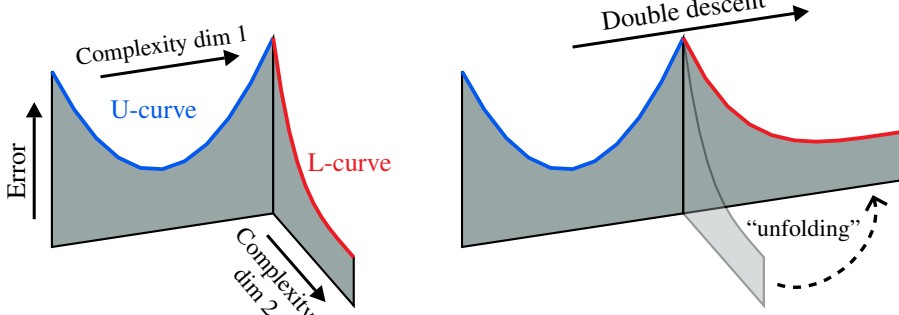

Figure 1: **A 3D generalization plot with two complexity axes unfolding into double descent.** A generalization plot with two complexity axes, each exhibiting a convex curve (left). By increasing raw parameters along different axes sequentially, a double descent effect appears to emerge along their composite axis (right).

The modern machine learning (ML) literature, conversely, focuses on far more flexible methods with relatively huge parameter counts and considers their generalization to *unseen* inputs [GBC16, Mur22]. A similar U-shaped curve was long accepted to also govern the complexity-generalization relationship of such methods [GBD92, Vap95, HTF09]– until highly overparametrized models, e.g. neural networks, were recently found to achieve near-zero training error *and* excellent test set performance [NTS14, BLLT20, Bel21]. In this light, the seminal paper of Belkin et al. (2019) [BHMM19] sparked a new line of research by arguing for a need to extend on the apparent limitations of classic understanding to account for a *double descent* in prediction performance as the total number of model parameters (and thus – presumably – model complexity) grows. This is illustrated in the right panel of Fig. 1. Intuitively, it is argued that while the traditional U-curve is appropriate for the regime in which the number of total model parameters $p$ is smaller than the number of instances $n$, it no longer holds in the modern, zero train-error, *interpolation regime* where $p > n$ – here, test error experiences a second descent. Further, it was demonstrated that this modern double descent view of model complexity applies not only in deep learning where it was first observed [BO96, NMB+18, SGd+18, ASS20], but also ubiquitously appears across many non-deep learning methods such as trees, boosting and even linear regression [BHMM19].

**Contributions.** In this work, we investigate whether the double descent behavior observed in recent empirical studies of such *non-deep* ML methods *truly* disagrees with the traditional notion of a U-shaped tradeoff between model complexity and prediction error. In two parts, we argue that once careful consideration is given to *what is being plotted* on the axes of these double descent plots, the originally counter-intuitive peaking behavior can be comprehensively explained under existing paradigms:

• **Part 1: Revisiting existing experimental evidence.** We show that in the experimental evidence for non-deep double descent – using trees, boosting, and linear regressions – there is implicitly *more than one complexity axis* along which the parameter count grows. Conceptually, as illustrated in Fig. 1, we demonstrate that this empirical evidence for double descent can thus be comprehensively explained as a consequence of an implicit *unfolding* of a 3D plot with two orthogonal complexity axes (that both individually display a classical convex curve) into a single 2D-curve. We also highlight that the location of the second descent is thus *not* inherently tied to the interpolation threshold. While this is straightforward to show for the tree- and boosting examples (Sec. 2), deconstructing the underlying axes in the linear regression example is non-trivial (and involves understanding the connections between min-norm solutions and *unsupervised dimensionality reduction*). Our analysis in this case (Sec. 3) could thus be of independent interest as a simple new interpretation of double descent in linear regression.

• **Part 2: Rethinking parameter counting through a classical statistics lens.** We then note that all methods considered in Part 1 can be interpreted as *smoothers* (Sec. 4), which are usually compared in terms of a measure of the *effective* (instead of raw) number of parameters they use when issuing predictions [HT90]. As existing measures were derived with *in-sample* prediction in mind, we propose a generalized effective parameter measure $p_{\hat{s}}^0$ that allows to consider arbitrary sets of inputs $\mathcal{I}_0$. Using $p_{\hat{s}}^0$ to measure complexity, we then indeed discover that the apparent double descent curves fold back into more traditional U-shapes – because $p_{\hat{s}}^0$ is *not actually increasing* in the interpolation regime. Further, we find that, in the interpolation regime, trained models tend to use a different number of effective parameters when issuing predictions on unseen test inputs than on previously observed training inputs. We also note that, while such interpolating models can generalize well to unseen inputs, overparametrization *cannot* improve their performance in terms of the in-sample prediction error originally of interest in statistics – providing a new reason for the historical absence of double descent curves. Finally, we discuss practical implications for e.g. model comparison.

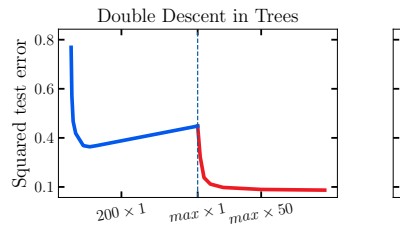 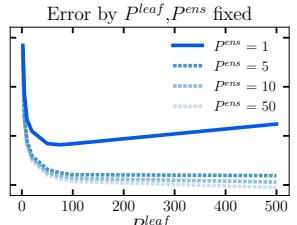 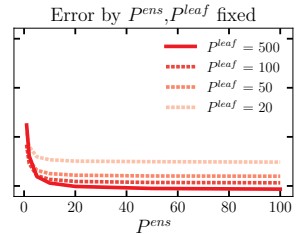

Figure 2: **Decomposing double descent for trees.** Reproducing [BHMM19]'s tree experiment (left). Test error by $P^{leaf}$ for fixed $P^{ens}$ (center). Test error by $P^{ens}$ for fixed $P^{leaf}$ (right).

## Part 1: Revisiting the evidence for double descent in non-deep ML models

**Experimental setup.** We center our study around the non-neural experiments in [BHMM19] as it is *the* seminal paper on double descent and provides the broadest account of non-deep learning methods that exhibit double descent. Through multiple empirical studies, [BHMM19] demonstrate that double descent arises in trees, boosting and linear regression. Below, we *re-analyze* these experiments and highlight that in each study, as we transition from the classical U-shaped regime into the subsequent second descent regime, *something else* implicitly changes in the model or training definition, fundamentally changing the class of models under consideration *exactly at the transition threshold* between the observed regimes – which is precisely the cause of the second descent phenomenon. In Sec. 2, we begin by investigating the tree and boosting experiments, where this is straightforward to show. In Sec. 3, we then investigate the linear regression example, where decomposing the underlying mechanisms is non-trivial. Throughout, we closely follow [BHMM19]'s experimental setup: they use standard benchmark datasets and train all ML methods by minimizing the *squared* loss[1]. Similarly to their work, we focus on results using MNIST (with $n_{train} = 10000$) in the main text. We present additional results, including other datasets, and further discussion of the experimental setup in Appendix E.

## 2 Warm-up: Observations of double descent in trees and boosting

### 2.1 Understanding double descent in trees

In the left panel of Fig. 2, we replicate the experiment in [BHMM19]'s Fig. 4, demonstrating double descent in trees. In their experiment, the number of model parameters is initially controlled through the maximum allowed number of terminal leaf nodes $P^{leaf}$. However, $P^{leaf}$ for a single tree cannot be increased past $n$ (which is when every leaf contains only one instance), and often $\max(P^{leaf}) < n$ whenever larger leaves are already pure. Therefore, when $P^{leaf}$ reaches its maximum, in order to further increase the raw number of parameters, it is necessary to change *how* further parameters are added to the model. [BHMM19] thus transition to showing how test error evolves as one averages over an increasing number $P^{ens}$ of different trees grown to full depth, where each tree will generally be distinct due to the randomness in features considered for each split. As one switches between plotting increasing $P^{leaf}$ and $P^{ens}$ on the x-axis, one is thus conceptually no longer increasing the number of parameters *within the same model class*: in fact, when $P^{ens} > 1$ one is no longer actually considering a tree, but instead *an ensemble* of trees (i.e. a random forest [Bre01] without bootstrapping).

In Fig. 2, we illustrate this empirically: in the center plot we show that, on the one hand, for fixed $P^{ens}$, error exhibits a classical convex U- (or L-)shape in tree-depth $P^{leaf}$. On the other hand, for fixed $P^{leaf}$ in the right plot, error also exhibits an L-shape in the number of trees $P^{ens}$, i.e. a convex shape without any ascent – which is in line with the known empirical observation that adding trees to a random forest generally does not hurt [HTF09, Ch. 15.3.4]. Thus, only by transitioning from increasing $P^{leaf}$ (with $P^{ens} = 1$) to increasing $P^{ens}$ (with $P^{leaf} = n$) – i.e. by connecting the two solid curves across the middle and the right plot – do we obtain the double

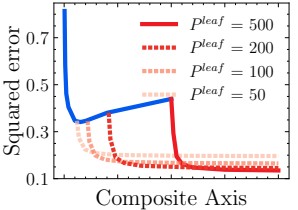

Figure 3: **Shifting the peak.** Transitioning from $P^{leaf}$ to $P^{ens}$ at different values of $P^{leaf}$.

---

[1]In multi-class settings, [BHMM19] use a one-vs-rest strategy and report squared loss summed across classes. Their use of the squared loss is supported by recent work on squared loss for classification [HB21, MNS+21], and practically implies the use of standard *regression* implementations of the considered ML methods.

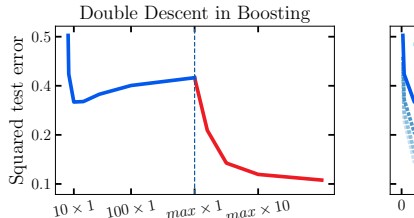 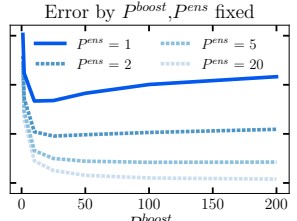 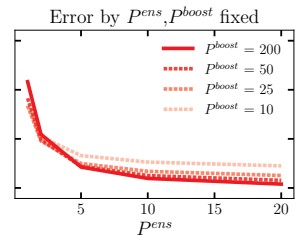

Figure 4: **Decomposing double descent for gradient boosting.** Reproducing [BHMM19]'s boosting experiment (left). Test error by $P^{boost}$ for fixed $P^{ens}$ (center). Test error by $P^{ens}$ for fixed $P^{boost}$ (right).

descent curve in the left plot of Fig. 2. In Fig. 3, we show that we could therefore *arbitrarily move or even remove* the first peak by changing *when* we switch from increasing parameters through $P^{leaf}$ to $P^{ens}$. Finally, we note that the interpolation threshold $p = n$ plays a special role only on the $P^{leaf}$ axis where it determines maximal depth, while parameters on the $P^{ens}$ axis can be increased indefinitely – further suggesting that parameter counts alone are not always meaningful[2].

## 2.2 Understanding double descent in gradient boosting

Another experiment is considered in Appendix S5 of [BHMM19] seeking to provide evidence for the emergence of double descent in gradient boosting. Recall that in gradient boosting, new base-learners (trees) are trained *sequentially*, accounting for current residuals by performing multiple *boosting rounds* which improve upon predictions of previous trees. In their experiments, [BHMM19] use trees with 10 leaves as base learners and a high learning rate of $\gamma = 0.85$ to encourage quick interpolation. The raw number of parameters is controlled by first increasing the number of boosting rounds $P^{boost}$ until the squared training error reaches approximately zero, after which $P^{boost}$ is fixed and ensembling of $P^{ens}$ independent models is used to further increase the raw parameter count.

In Fig. 4, we first replicate the original experiment and then again provide experiments varying each of $P^{boost}$ and $P^{ens}$ separately. Our findings parallel those above for trees: for a fixed number of ensemble members $P^{ens}$, test error has a U- or L-shape in the number of boosting rounds $P^{boost}$ and an L-shape in $P^{ens}$ for fixed boosting rounds $P^{boost}$. As a consequence, a double descent shape occurs *only* when and where we switch from one method of increasing complexity to another.

## 3 Deep dive: Understanding double descent in linear regression

We are now ready to consider Fig. 2 of [BHMM19], which provides experiments demonstrating double descent in the case of linear regression. Recall that linear regression with $\mathbf{y} \in \mathbb{R}^n$ and $\mathbf{X} \in \mathbb{R}^{n \times d}$ estimates the coefficients in a model $\mathbf{y} = \mathbf{X}\boldsymbol{\beta}$, thus the number of raw model parameters equals the number of input dimensions (i.e. the dimension $d$ of the regression coefficient vector $\boldsymbol{\beta}$) by design. Therefore, in order to flexibly control the number of model parameters, [BHMM19] apply basis expansions using random Fourier features (RFF). Specifically, given input $\mathbf{x} \in \mathbb{R}^d$, the number of raw model parameters $P^\phi$ is controlled by randomly generating features $\phi_p(\mathbf{x}) = \text{Re}(\exp^{\sqrt{-1}\mathbf{v}_p^T \mathbf{x}})$ for all $p \leq P^\phi$, where each $\mathbf{v}_p \overset{\text{iid}}{\sim} \mathcal{N}(\mathbf{0}, \frac{1}{5^2} \cdot \mathbf{I}_d)$. For any given number of features $P^\phi$, these are stacked to give a $n \times P^\phi$ dimensional random design matrix $\boldsymbol{\Phi}$, which is then used to solve the regression problem $\mathbf{y} = \boldsymbol{\Phi}\boldsymbol{\beta}$ by least squares. For $P^\phi \leq n$, this has a unique solution ($\hat{\boldsymbol{\beta}} = (\boldsymbol{\Phi}^T\boldsymbol{\Phi})^{-1}\boldsymbol{\Phi}^T\mathbf{y}$) while for $P^\phi > n$ the problem becomes underdetermined (i.e. there are infinite solutions) which is why [BHMM19] rely on a specific choice: the min-norm solution ($\hat{\boldsymbol{\beta}} = \boldsymbol{\Phi}^T(\boldsymbol{\Phi}\boldsymbol{\Phi}^T)^{-1}\mathbf{y}$).

Unlike the experiments discussed in Sec. 2, there appears to be only one obvious mechanism for increasing raw parameters in this case study. Instead, as we show in Sec. 3.1, the change in the used solution at $P^\phi = n$ turns out to be the crucial factor here: we find that the min-norm solution leads to implicit *unsupervised dimensionality reduction*, resulting in two distinct mechanisms for increasing the total parameter count in linear regression. Then, we again demonstrate empirically in Sec. 3.2 that each individual mechanism is indeed associated with a standard generalization curve, such that the combined generalization curve exhibits double descent only because they are applied in succession.

---

[2]A related point is raised in [BM21], who notice that a double descent phenomenon in random forests appears in [BHMM19] only because the *total* number of leaves is placed on the x-axis, while the true complexity of a forest is actually better characterized by the *average* number of leaves in its trees according to results in learning theory. However, this complexity measure clearly does not change once $P^{leaf}$ is fixed in the original experiment.

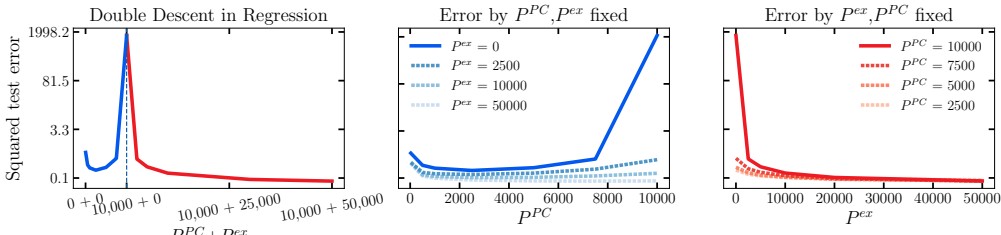

Figure 5: **Decomposing double descent for RFF Regression.** Double descent reproduced from [BHMM19] (left) can be decomposed into the standard U-curve of ordinary linear regression with $P^{PC}$ features (center) and decreasing error achieved by a *fixed capacity* model with basis improving in $P^{ex}$ (right).

### 3.1 Understanding the connections between min-norm solutions and dimensionality reduction

In this section, we show that, while the min-norm solution finds coefficients $\hat{\boldsymbol{\beta}}$ of *raw* dimension $P^{\phi}$, only $n$ of its dimensions are well-determined – i.e. the true parameter count is not actually increasing in $P^{\phi}$ once $P^{\phi} > n$. Conceptually, this is because min-norm solutions project $\mathbf{y}$ onto the row-space of $\boldsymbol{\Phi}$, which is $n-$dimensional as $rank(\boldsymbol{\Phi}) = \min(P^{\phi}, n)$ (when $\boldsymbol{\Phi}$ has full rank). To make the consequence of this more explicit, we can show that the min-norm solution (which has $P^{\phi}$ raw parameters) can always be represented by using a $n$-dimensional coefficient vector applied to a $n-$dimensional basis of $\boldsymbol{\Phi}$. In fact, as we formalize in Proposition 1, this becomes most salient when noting that applying the min-norm solution to $\boldsymbol{\Phi}$ is *exactly* equivalent to a learning algorithm that (i) first constructs a $n-$dimensional basis $\mathbf{B}_{SVD}$ from the $n$ right singular vectors of the input matrix computed using the singular value decomposition (SVD) in an *unsupervised pre-processing step* and (ii) then applies standard (fully determined) least squares using the discovered $n-$dimensional basis[3].

**Proposition 1.** *[Min-norm least squares as dimensionality reduction.] For a full rank matrix $\mathbf{X} \in \mathbb{R}^{n \times d}$ with $n < d$ and a vector of targets $\mathbf{y} \in \mathbb{R}^n$, the min-norm least squares solution $\hat{\boldsymbol{\beta}}^{MN} = \{\min_{\boldsymbol{\beta}} ||\boldsymbol{\beta}||_2^2 : \mathbf{X}\boldsymbol{\beta} = \mathbf{y}\}$ and the least squares solution $\hat{\boldsymbol{\beta}}^{SVD} = \{\boldsymbol{\beta} : \mathbf{B}\boldsymbol{\beta} = \mathbf{y}\}$ using the matrix of basis vectors $\mathbf{B} \in \mathbb{R}^{n \times n}$, constructed using the first $n$ right singular vectors of X, are equivalent; i.e. $\mathbf{x}^T \hat{\boldsymbol{\beta}}^{MN} = \mathbf{b}^T \hat{\boldsymbol{\beta}}^{SVD}$ for all $\mathbf{x} \in \mathbb{R}^d$ and corresponding basis representation $\mathbf{b} \equiv \mathbf{b}(\mathbf{x})$.*
*Proof.* Please refer to Appendix B.2. $\square$

Then what is *really* causing the second descent if not an increasing number of fitted dimensions? While the addition of feature dimensions *does* correspond to an increase in fitted model parameters while $P^{\phi} < n$, the performance gains in the $P^{\phi} > n$ regime are better explained as a linear model of fixed size $n$ being fit to an *increasingly rich basis constructed in an unsupervised step*. To disentangle the two mechanisms further, we note that the procedure described above, when applied to a centered design matrix[4] is a special case of principal component (PC) regression [Jol82], where we select *all* empirical principal components to form a complete basis of $\boldsymbol{\Phi}$. The more general approach would instead consist of selecting the top $P^{PC}$ principal components and fitting a linear model to that basis. Varying $P^{PC}$, the number of used principal components, is thus actually the first mechanism by which the raw parameter count can be altered; this controls the number of parameters being fit in the supervised step. The second, less obvious, mechanism is then the number of *excess features* $P^{ex} = P^{\phi} - P^{PC}$; this is the number of raw dimensions that only contribute to the creation of a richer basis, which is learned in an unsupervised manner[5].

Based on this, we can now provide a new explanation for the emergence of double descent in this case study: due to the use of the min-norm solution, the two uncovered mechanisms are implicitly entangled

---

[3]This application of the fundamental connection between min-norm solutions and the singular value decomposition (see e.g. [GVL13, Ch. 5.7]) reinforces previous works which have noted related links between the min-norm solution and dimensionality reduction [RD98, KL16].

[4]Centering reduces the rank of the input matrix by 1 and thus requires appending an intercept to the PC design matrix. Without centering, the procedure is using the so-called *uncentered* PCs [CJ09] instead. We use the centered version with intercept in our experiments, but obtained identical results when using uncentered PCs.

[5]The *implicit inductive bias* encoded in this step essentially consists of constructing and choosing the top-$P^{PC}$ features that capture the *directions of maximum variation* in the data. Within this inductive bias, the role of $P^{ex}$ appears to be that – as more excess features are added – the variation captured by each of the top-$P^{PC}$ PCs is likely to increase. Using the directions of maximum variation is certainly not guaranteed to be optimal [Jol82], but it tends to be an effective inductive bias in practice as noted by Tukey [Tuk77] who suggested that high variance components are likely to be more important for prediction unless nature is "downright mean".

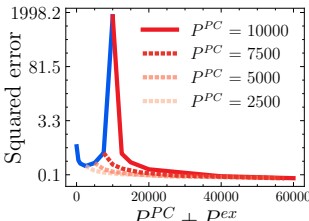
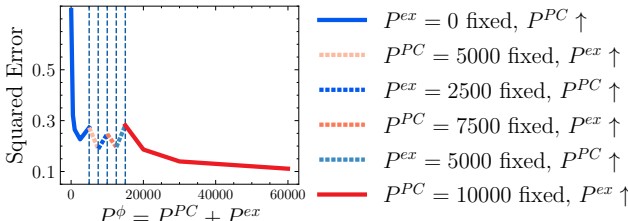

(a) **Shifting the peak.** Transitioning from $P^{PC}$ to $P^{ex}$ at different values of $P^{PC}$.

(b) **Multiple descent.** Generalization curves with arbitrarily many peaks and locations (including peaks at $P^\phi > n$) can be created by switching between increasing parameters through $P^{PC}$ and $P^{ex}$ multiple times.

Figure 6: **Disentangling double descent from the interpolation threshold.** The location of the peak(s) in RFF regression generalization error is not inherently linked to the point where $P^\phi = n$. Instead, changes in the mechanism for parameter increase determine the appearance of peaks.

through $P^\phi = P^{PC} + P^{ex}$, $P^{PC} = \min(n, P^\phi)$ and $P^{ex} = \max(0, P^\phi - n)$. Thus, we have indeed arrived back at a setup that parallels the previous two experiments: when $P^\phi \leq n$, $P^{PC}$ increases monotonically while $P^{ex} = 0$ is constant, while when $P^\phi > n$ we have constant $P^{PC} = n$ but $P^{ex}$ increases monotonically. Below, we can now test empirically whether studying the two mechanisms separately indeed leads us back to standard convex curves as before. In particular, we also show that – while a transition between the mechanisms increasing $P^{PC}$ and $P^{ex}$ naturally happens at $P^\phi = n$ in the original experiments – it is possible to transition elsewhere across the implied complexity axes, creating other thresholds, and to thus disentangle the double descent phenomenon from $n$.

## 3.2 Empirical resolutions to double descent in RFF regression

Mirroring the analyses in Sec. 2, we now investigate the effects of $P^{PC}$ and $P^{ex}$ in Fig. 5. In the left plot, we once more replicate [BHMM19]'s original experiment, and observe the same apparent trend that double descent emerges as we increase the number of raw parameters $P^\phi = P^{PC} + P^{ex}$ (where the min-norm solution needs to be applied once $P^\phi = n$). We then proceed to analyze the effects of varying $P^{PC}$ and $P^{ex}$ separately (while holding the other fixed). As before, we observe in the center plot that varying $P^{PC}$ (determining the actual number of parameters being fit in the regression) for different levels of excess features indeed gives rise to the traditional U-shaped generalization curve. Conversely, in the right plot, we observe that increasing $P^{ex}$ for a fixed number of $P^{PC}$ results in an L-shaped generalization curve – indeed providing evidence that the effect of increasing the number of raw parameters past $n$ in the original experiment can be more accurately explained as a gradual improvement in the quality of a basis to which a *fixed capacity model* is being fit.

As before, note that if we connect the solid lines in the center and right plots we recover exactly the double descent curve shown in the left plot of Fig. 5. Alternatively, as we demonstrate in Fig. 6(a), fixing $P^{PC}$ at other values and then starting to increase the total number of parameters through $P^{ex}$ allows us *to move or remove the first peak arbitrarily*. In Fig. 6(b), we demonstrate that one could even create multiple peaks[6] by switching between parameter-increasing mechanisms *more than once*. This highlights that, while the transition from parameter increase through $P^{PC}$ to $P^{ex}$ naturally occurs at $P^\phi = n$ due to the use of the min-norm solution, the second descent is not actually caused by the interpolation threshold $P^\phi = n$ itself – but rather is due to the implicit change in model at exactly this point. Indeed, comparing the generalization curves in Fig. 6 with their train-error trajectories which we plot in Appendix E.2, it becomes clear that such a second descent can also occur in models that have not yet and will never achieve interpolation of the training data.

## 4 Part 2: Rethinking parameter counting through a classical statistics lens

Thus far, we have highlighted that "not all model parameters are created equal" – i.e. the intuitive notion that not all ways of increasing the number of *raw* parameters in an ML method have the same

---

[6]This experiment is inspired by [CMBK21], who show *multiple* descent in regression by controlling the *data-generating process* (DGP), altering the order of revealing new (un)informative features. Even more striking than through changing the DGP, we show that simply reordering the mechanisms by which *the same raw parameters* are added to the model allows us to arbitrarily increase and decrease test loss.

effect. However, as we saw in the linear regression example, it is not always trivial to deconstruct the underlying mechanisms driving performance. Therefore, instead of having to reason about implicit complexity axes on a case-by-case basis for different models, hyperparameters, or inductive biases, we would rather be able to *quantify* the effect of these factors objectively. In what follows, we highlight that all previously considered methods can be interpreted as *smoothers* (in the classical statistics sense [HT90]). By making this connection in Sec. 4.1, we can exploit the properties of this class of models providing us with measures of their *effective* number of parameters. After adapting this concept to our setting (Sec. 4.2), we are finally able to *re-calibrate* the complexity axis of the original double descent experiments, finding that they do indeed fold back into more traditional U-shapes (Sec. 4.3).

## 4.1 Connections to smoothers

Smoothers are a class of supervised learning methods that summarize the relationship between outcomes $Y \in \mathcal{Y} \subset \mathbb{R}^k$ and inputs $X \in \mathcal{X} \subset \mathbb{R}^d$ by "smoothing" over values of $Y$ observed in training. More formally, let $k = 1$ w.l.o.g., and denote by $\mathcal{D}^{\text{train}} = \{(y_i, x_i)\}_{i=1}^n$ the training realizations of $(X, Y) \in \mathcal{X} \times \mathcal{Y}$ and by $\mathbf{y}_{\text{train}} = (y_1, \ldots, y_n)^T$ the $n \times 1$ vector of training outcomes with respective training indices $\mathcal{I}_{\text{train}} = \{1, \ldots, n\}$. Then, for any admissible input $x_0 \in \mathcal{X}$, a smoother issues predictions

$$\hat{f}(x_0) = \hat{\mathbf{s}}(x_0)\mathbf{y}_{\text{train}} = \sum_{i \in \mathcal{I}_{\text{train}}} \hat{s}^i(x_0)y_i \tag{1}$$

where $\hat{\mathbf{s}}(x_0) = (\hat{s}^1(x_0), \ldots, \hat{s}^n(x_0))$ is a $1 \times n$ vector containing smoother weights for input $x_0$. A smoother is *linear* if $\hat{\mathbf{s}}(\cdot)$ does not depend on $\mathbf{y}_{\text{train}}$. The most well-known examples of smoothers rely on weighted (moving-) averages, which includes k-nearest neighbor (kNN) methods and kernel smoothers as special cases. (Local) linear regression and basis-expanded linear regressions, including splines, are other popular examples of linear smoothers (see e.g. [HT90, Ch. 2-3]).

In Appendix C, we show that all methods studied in Part 1 can be interpreted as smoothers, and derive $\hat{\mathbf{s}}(\cdot)$ for each method. To provide some intuition, note that linear regression is a simple textbook example of a linear smoother [HT90], where $\hat{\mathbf{s}}(\cdot)$ is constructed from the so-called projection (or hat) matrix [Eub84]. Further, trees – which issue predictions by averaging training outcomes within leaves – are sometimes interpreted as *adaptive nearest neighbor methods* [HTF09, Ch. 15.4.3] with *learned* (i.e. non-linear) weights, and as a corollary, boosted trees and sums of either admit similar interpretations.

## 4.2 A generalized measure of the *effective* number of parameters used by a smoother

The *effective number of parameters $p_e$* of a smoother was introduced to provide a measure of model complexity which can account for a broad class of models as well as different levels of model regularization (see e.g. [HT90, Ch. 3.5], [HTF09, Ch. 7.6]). This generalized the approach of simply counting raw parameters – which is not always possible or appropriate – to measure model complexity. This concept is *calibrated* towards linear regression so that, as we might desire, effective and raw parameter numbers are equal in the case of ordinary linear regression with $p < n$. In this section, we adapt the *variance based* effective parameter definition discussed in [HT90, Ch. 3.5]. Because for fixed $\hat{\mathbf{s}}(\cdot)$ and outcomes generated with homoskedastic variance $\sigma^2$ we have $Var(\hat{f}(x_0)) = ||\hat{\mathbf{s}}(x_0)||^2\sigma^2$, this definition uses that $\frac{1}{n}\sum_{i \in \mathcal{I}_{\text{train}}} Var(\hat{f}(x_i)) = \frac{\sigma^2}{n}\sum_{i \in \mathcal{I}_{\text{train}}} ||\hat{\mathbf{s}}(x_i)||^2 = \frac{\sigma^2}{n}p$ for linear regression: one can define $p_e = \sum_{i \in \mathcal{I}_{\text{train}}} ||\hat{\mathbf{s}}(x_i)||^2$ and thus have $p_e = p$ for ordinary linear regression with $n < p$. As we discuss further in Appendix D, other variations of such effective parameter count definitions can also be found in the literature. However, this choice is particularly appropriate for our purposes as it has the unique characteristic that it can easily be *adapted to arbitrary input points* – a key distinction we will motivate next.

Historically, the smoothing literature has primarily focused on prediction in the classical fixed design setup where expected in-sample prediction error on the *training inputs $x_i$*, with only newly sampled targets $y_i'$, was considered the main quantity of interest [HT90, RT19]. The modern ML literature, on the other hand, largely focuses its evaluations on out-of-sample prediction error in which we are interested in model performance, or *generalization*, on both unseen targets *and* unseen inputs (see e.g. [GBC16, Ch. 5.2]; [Mur22, Ch. 4.1]). To make effective parameters fit for modern purposes, it is therefore necessary to adapt $p_e$ to measure the level of smoothing applied *conditional on a given input*, thus distinguishing between training and testing inputs. As $||\hat{\mathbf{s}}(x_0)||^2$ can be computed for *any* input $x_0$, this is straightforward and can be done by replacing $\mathcal{I}_{\text{train}}$ in the definition of $p_e$ by any other set of inputs indexed by $\mathcal{I}_0$. Note that the scale of $p_e$ would then depend on $|\mathcal{I}_0|$ due to the summation,

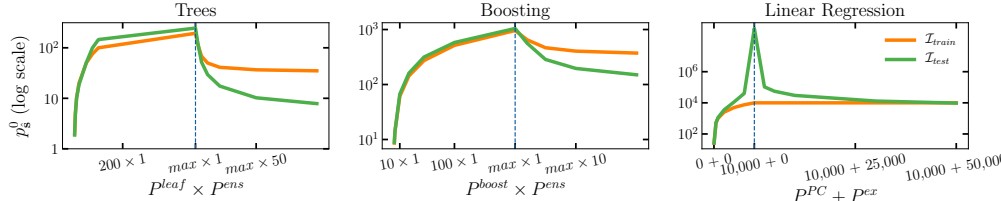

**Figure 7: The effective number of parameters does not increase past the transition threshold.**
Plotting $p_{\hat{\mathbf{s}}}^{\text{train}}$ (orange) and $p_{\hat{\mathbf{s}}}^{\text{test}}$ (green) for the tree (left), boosting (centre) and RFF-linear regression (right) experiments considered in Part 1, using the original composite parameter axes of [BHMM19].

while it should actually depend on the number of training examples $n$ (previously implicitly captured through $|\mathcal{I}_{\text{train}}|$) across which the smoothing occurs – thus we also need to recalibrate our definition by $n/|\mathcal{I}_0|$. As presented in Definition 1, we can then measure the *generalized* effective number of parameters $p_{\hat{\mathbf{s}}}^0$ used by a smoother when issuing predictions for *any* set of inputs $\mathcal{I}_0$.

**Definition 1** (Generalized Effective Number of Parameters). *For a set of inputs $\{x_j^0\}_{j \in \mathcal{I}_0}$, define the Generalized Effective Number of Parameters $p_{\hat{\mathbf{s}}}^0$ used by a smoother with weights $\hat{\mathbf{s}}(\cdot)$ as*

$$p_{\hat{\mathbf{s}}}^0 \equiv p(\mathcal{I}_0, \hat{\mathbf{s}}(\cdot)) = \frac{n}{|\mathcal{I}_0|} \sum_{j \in \mathcal{I}_0} ||\hat{\mathbf{s}}(x_j^0)||^2 \tag{2}$$

Note that for the training inputs we recover the original quantity exactly ($p_e = p_{\hat{\mathbf{s}}}^{\text{train}}$). Further, for moving-average smoothers with $\sum_{i=1}^n \hat{s}^i(x_0) = 1$ and $0 \le \hat{s}^i(x_0) \le 1$ for all $i$, it holds that $1 \le p_{\hat{\mathbf{s}}}^0 \le n$. Finally, for kNN estimators we have $p_{\hat{\mathbf{s}}}^0 = \frac{n}{k}$, so that for smoothers satisfying $\sum_{i=1}^n \hat{s}^i(x_0) = 1$ and $0 \le \hat{s}^i(x_0) \le 1$ for all $i$, the quantity $\tilde{k}_{\hat{\mathbf{s}}}^0 = \frac{n}{p_{\hat{\mathbf{s}}}^0}$ also admits an interesting interpretation as measuring the *effective number of nearest neighbors*.

### 4.3 Back to U: Measuring *effective* parameters folds apparent double descent curves

Using Definition 1, we can now measure the *effective* number of parameters to re-examine the examples of double descent described in Sections 2 & 3. We plot the results of the 0-vs-all sub-problem in this section as the 10 one-vs-all models in the full experiment are each individually endowed with effective parameter counts $p_{\hat{\mathbf{s}}}^0$. Fig. 7 presents results plotting this quantity measured on the training inputs ($p_{\hat{\mathbf{s}}}^{\text{train}}$) and testing inputs ($p_{\hat{\mathbf{s}}}^{\text{test}}$) against the original, composite parameter axes of [BHMM19], Fig. 8 considers the behavior of $p_{\hat{\mathbf{s}}}^{\text{test}}$ for the two distinct mechanisms of parameter increase separately, and in Fig. 9 we plot test error against $p_{\hat{\mathbf{s}}}^{\text{test}}$, finally replacing raw with effective parameter axes.

By examining the behavior of $p_{\hat{\mathbf{s}}}^0$ in Fig. 7 and Fig. 8, several interesting insights emerge. First, we observe that these results complement the intuition developed in Part 1: In all cases, measures of the *effective* number of parameters never increase after the threshold where the mechanism for increasing the raw parameter count changes. Second, the distinction of measuring effective number of parameters used on *unseen inputs*, i.e. using $p_{\hat{\mathbf{s}}}^{\text{test}}$ instead of $p_{\hat{\mathbf{s}}}^{\text{train}}$, indeed better tracks the double descent phenomenon which itself emerges in generalization error (i.e. is also estimated on unseen inputs). This is best illustrated in Fig. 7 in the linear regression example where $p_{\hat{\mathbf{s}}}^{\text{train}} = n$ is *constant* once $P^\phi \ge n$ – as is expected: simple matrix algebra reveals that $\hat{\mathbf{s}}(x_i) = \mathbf{e}_i$ (with $\mathbf{e}_i$ the $i^{th}$ indicator vector) for any training example $x_i$ once $P^\phi \ge n$. Intuitively, no smoothing across training labels occurs on the training inputs after the interpolation threshold as each input simply predicts its own label (note that this would also be the case for regression trees if each training example fell into its own leaf; this does not occur in these experiments as leaves are already pure before this point). As we

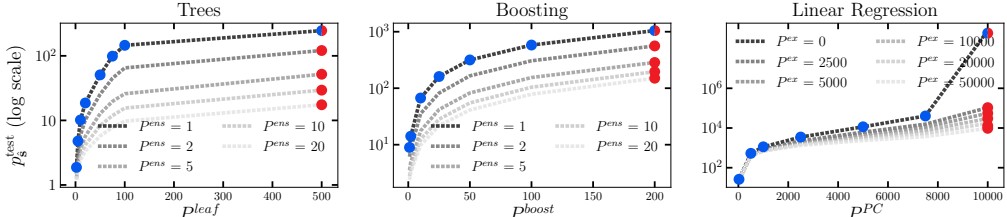

**Figure 8: Understanding the behavior of the test-time effective number of parameters across the two parameter axes of each ML method.** Plotting $p_{\hat{\mathbf{s}}}^{\text{test}}$ by $P^{leaf}$ for fixed $P^{ens}$ in trees (left), by $P^{boost}$ for fixed $P^{ens}$ in boosting (centre) and by $P^{PC}$ for fixed $P^{ex}$ in RFF-linear regression (right) highlights that increases along the first parameter axes increase $p_{\hat{\mathbf{s}}}^{\text{test}}$, while increases along the second axes *decrease* $p_{\hat{\mathbf{s}}}^{\text{test}}$. Larger blue and red points (•, •) are points from [BHMM19]'s original parameter sequence.

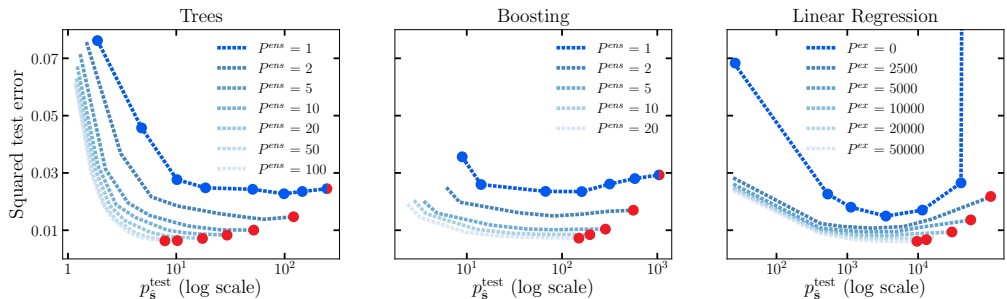

Figure 9: **Back to U.** Plotting test error against the *effective* number of parameters as measured by $p_{\hat{\mathbf{s}}}^{\text{test}}$ (larger blue and red points: ●, ●) for trees (left), boosting (centre) and RFF-linear regression (right) eliminates the double descent shape. In fact, points from the first dimension of [BHMM19]'s composite axis (●) continue to produce the familiar U-shape, while points from the second dimension of the composite axis (●) – which originally created the apparent second descent – *fold* back into classical U-shapes. Dotted lines (▪▪▪) show the effect of perturbing the first complexity parameter at different fixed values of the second.

make more precise in Appendix C.2, this observation leads us to an interesting impossibility result: the *in-sample* prediction error of *interpolating models* therefore cannot experience a second descent in the (raw) overparameterized regime. This provides a new reason for the historical absence of double descent shapes (in one of *the* primary settings in which the U-shaped tradeoff was originally motivated [HT90]), complementing the discussion on the lack of previous observations in [BHMM19]. Third, while $p_{\hat{\mathbf{s}}}^{\text{train}}$ is therefore not useful to understand double descent in generalization error, we note that for *unseen inputs* different levels of smoothing across the training labels can occur even in the interpolation regime – and this, as quantified by $p_{\hat{\mathbf{s}}}^{\text{test}}$, *does* result in an informative complexity proxy.

This becomes obvious in Fig. 9, where we plot the test errors of [BHMM19]'s experimental setup against $p_{\hat{\mathbf{s}}}^{\text{test}}$ (larger blue and red points: ●, ●). For each such point after the interpolation threshold (●), we also provide dotted contour lines (▪▪▪) representing the effect of reducing the parameters along the first complexity axis ($P^{leaf}$, $P^{boost}$ and $P^{PC}$), resulting in models that can no longer *perfectly interpolate* (e.g. for trees, we fix their number $P^{ens}$ and reduce their depth through $P^{leaf}$). Finally, we discover that once we control for the effective parameters of each model in this way, we consistently observe that each example indeed folds back into shapes that are best characterized as standard convex (U-shaped) curves – providing a resolution to the apparent tension between traditional statistical intuition and double descent! We also note an interesting phenomenon: increasing $P^{ens}$ and $P^{ex}$ appears to not only shift the complexity-generalization contours downwards (decreasing error for each value of $P^{leaf}$, $P^{boost}$ and $P^{PC}$), but also shifts them to the left (decreasing the effective complexity implied by each value of $P^{leaf}$, $P^{boost}$ and $P^{PC}$, as also highlighted in Fig. 8) and flattens their ascent (making error less sensitive to interpolation). That is, models in (or close to) the interpolation regime discovered by [BHMM19] exhibit a very interesting behavior where setting $P^{ex}$ and $P^{ens}$ larger actually decreases the test error through a *reduction* in the effective number of parameters used.

## 5 Consolidation with Related Work

In this section, we consolidate our findings with the most relevant recent work on *understanding* double descent; we provide an additional broader review of the literature in Appendix A. A large portion of this literature has focused on modeling double descent in the *raw* number of parameters in linear regression, where these parameter counts are varied by employing random feature models [BHMM19, BHX20] or by varying the ratio of $n$ to input feature dimension $d$ [ASS20, BLLT20, DLM20, HMRT22]. This literature, typically studying the behavior and conditioning of feature covariance matrices, has produced precise theoretical analyses of double descent for particular models as the *raw* number of parameters is increased (e.g. [BLLT20, HMRT22]). Our insights in Sections 3 and 4 are *complementary* to this line of work, providing a new perspective in terms of (a) decomposing *raw* parameters into two separate complexity axes which jointly produce the double descent shape only due to the choice of (min-norm) solution and (b) the underlying *effective* number of parameters used by a model. Additionally, using theoretical insights from this line of work can make more precise the role of $P^{ex}$ in improving the quality of a learned basis: as we show in Appendix B.3, increasing $P^{ex}$ in our experiments indeed leads to the PC feature matrices being better conditioned.

Further, unrelated to the smoothing lens we use in Part 2 of this paper, different notions of effective parameters or complexity have appeared in other studies of double descent in specific

ML methods: [BM21] use Rademacher complexity (RC) in their study of random forests but find that RC cannot explain their generalization because forests do not have lower RC than individual trees (unlike the behavior of our $p_{\hat{\mathbf{s}}}^{\text{test}}$ in this case). [MBW20] use [Mac91]'s Bayesian interpretation of effective parameters based on the Hessian of the training loss when studying neural networks. While this proxy was originally motivated in a linear regression setting, it *cannot* explain double descent in linear regression because, as discussed further in Appendix D, it does not decrease therein once $p > n$. Finally, [DLM20] compute effective ridge penalties implied by min-norm linear regression, and [DSYW20] consider minimum description length principles to measure complexity in ridge regression. Relative to this literature, our work differs not only in terms of the ML methods we can consider (in particular, we are uniquely able to explain the tree- and boosting experiments through our lens), but also in the insights we provide e.g. we distinctively propose to distinguish between effective parameters used on train- versus test-examples, which we showed to be crucial to explaining the double descent phenomenon in Sec. 4.3.

## 6  Conclusion and Discussion

**Conclusion:** *A Resolution to the ostensible tension between non-deep double descent and statistical intuition.* We demonstrated that existing experimental evidence for double descent in trees, boosting and linear regression does not contradict the traditional notion of a U-shaped complexity-generalization curve: to the contrary, we showed that in all three cases, there are actually two *independent* underlying complexity axes that each exhibit a standard convex shape, and that the observed double descent phenomenon is a direct consequence of transitioning between these two distinct mechanisms of increasing the total number of model parameters. Furthermore, we highlighted that when we plot a measure of the *effective, test-time,* parameter count (instead of *raw* parameters) on their x-axes, the apparent double descent curves indeed fold back into more traditional U-shapes.

**What about *deep* double descent?** In this work, we intentionally limited ourselves to the study of *non-deep* double descent. Whether the approach pursued in this work could provide an alternative path to understanding double descent in the case of deep learning – arguably its most prominent setting – is thus a very natural next question. It may indeed be instructive to investigate whether there also exist multiple implicitly entangled complexity axes in neural networks, and whether this may help to explain double descent in that setting. In particular, one promising approach to bridging this gap could be to combine our insights of Sec. 3 with the known connections between random feature models and two-layer neural networks [SGT18], and stochastic gradient descent and min-norm solutions [GWB+17]. We consider this a fruitful and non-trivial direction for future research.

**What are the practical implications of these findings?** On the one hand, with regards to the specific ML methods under investigation, our empirical results in Sections 2 and 3 imply interesting trade-offs between the need for hyperparameter tuning and raw model size. All methods appear to have one hyperparameter axis to which error can be highly sensitive – $P^{leaf}$, $P^{boost}$ and $P^{PC}$ – while along the other axis, "bigger is better" (or at least, not worse). In fact, it appears that the higher $P^{ens}$ or $P^{ex}$, the less sensitive the model becomes to changes along the first axis. This may constitute anecdotal evidence that the respective first axis can be best understood as a train-time bias-reduction axis – it controls how well the *training* data can be fit (increasing parameters along this axis reduces underfitting – only when set to its maximum can interpolation be achieved). The second axis, conversely, appears to predominantly achieve variance-reduction at *test-time*: it decreases $||\hat{\mathbf{s}}(x_0)||$, reducing the impact of noise by smoothing over more training examples when issuing predictions for unseen inputs.

On the other hand, our results in Sec. 4 suggest interesting new avenues for model selection more generally, by highlighting potential routes of redemption for classical criteria trading off in-sample performance and parameter counts (e.g. [Aka74, Mal73]) – which have been largely abandoned in ML in favor of selection strategies evaluating held-out prediction error [Ras18]. While criteria based on *raw* parameter counts may be outdated in the modern ML regime, selection criteria based on *effective* parameter counts $p_{\hat{\mathbf{s}}}^{\text{test}}$ used on a test-set could provide an interesting new alternative, with the advantage of not requiring access to labels on held-out data, unlike error-based methods. In Appendix E.5, we provide anecdotal evidence that when choosing between models with different hyperparameter settings that all achieve *zero training error*, considering each model's $p_{\hat{\mathbf{s}}}^{\text{test}}$ could be used to identify a good choice in terms of generalization performance: we illustrate this for the case of gradient boosting where we use $p_{\hat{\mathbf{s}}}^{\text{test}}$ to choose additional hyperparameters. Investigating such approaches to model selection more extensively could be another promising avenue for future work.

## Acknowledgements

We would like to thank Fergus Imrie, Tennison Liu, James Fox, and the anonymous reviewers for insightful comments and discussions on earlier drafts of this paper. AC and AJ gratefully acknowledge funding from AstraZeneca and the Cystic Fibrosis Trust respectively. This work was supported by Azure sponsorship credits granted by Microsoft's AI for Good Research Lab.

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

# Appendix

This Appendix is structured as follows: In Appendix A, we present an additional, broader, literature review. In Appendix B, we present additional background on the linear regression case study, including the proof of Proposition 1 (Appendix B.2) and an empirical study of the effect of excess features on the conditioning of PC feature matrices (Appendix B.3). In Appendix C, we present additional background on smoothers, including an analysis of the impossibility of double descent in prediction error in fixed design settings (Appendix C.2) and derivation of the smoothing matrices associated with linear regression, trees and boosting (Appendix C.3). In Appendix D, we present additional background and other definitions of effective parameter measures. Finally, in Appendix E, we further discuss the experimental setup and close by presenting additional results.

## A  Additional literature review

**Double descent as a phenomenon.** Although double descent only gained popular attention since [BHMM19]'s influential paper on the topic, the phenomenon itself had previously been observed: [LVM$^+$20] provide a historical note highlighting that [VCR89] may have been the first to demonstrate double descent in min-norm linear regression; [BHMM19] themselves note that double descent-like phase transitions in neural networks had previously been observed in [BO96, ASS20, NMB$^+$18, SGd$^+$18]. As discussed above, in addition to linear regressions and neural networks, [BHMM19] also present double descent curves for trees and boosting. Since then, a rich literature on other deep double descent phenomena has emerged, including double descent in the number of training epochs[NKB$^+$21], sparsity [HXZQ22], data generation [LDB21] and transfer learning [DB22]. We also note that the occurrence of double descent has been studied for subspace regression methods themselves [DMLB20, THV22], but has not been linked to double descent in min-norm linear regressions as we do here. For a recent review of double descent more broadly see [VL22].

**Theoretical understanding of double descent.** In addition to the studies of double descent in linear regression discussed in the main text[ASS20, BLLT20, DLM20, BHX20, KSR$^+$21, HMRT22], we note that further theoretical studies of double descent in neural networks exist (see e.g. [DMB21] for a comprehensive review). These mainly focus on exact expressions of bias and variance terms by taking into account all sources of randomness in model training and data sampling [NMB$^+$18, AP20, dRBK20, LD21]. Finally, different to our peak-moving experiments in Sec. 3.2 and Appendix E.2 which show that the location of the second descent in all original non-deep experiments of [BHMM19] is determined by a change in the underlying method or training procedure, [CMBK21] show that one can "design your own generalization curve" in linear regression by controlling the data-generating process through the order by which newly (un)informative features are revealed, meaning that one could construct a learning curve with essentially arbitrary number of descents in this way. Similarly, we show in Fig. 6(b) in the main text and Fig. 13 in Appendix E.2 that multiple peaks can be achieved by switching between mechanisms for adding parameters multiple times. Note that [CMBK21] consider only the effect of data-generating mechanisms and arrive at the conclusion that the generalization curves observed in practice must arise due to interactions between typical data-generating processes and inductive biases of algorithms and "highlight that the nature of these interactions is far from understood" [CMBK21, p. 3] – with our paper, we contributed to the understanding of these interactions by studying the role of *changes* in inductive biases in the case of trees, boosting and linear regression.

**Interpolation regime.** Entangled in the double descent literature is the *interpolation regime* or the *benign overfitting* effect (e.g. [BMM18, MBB18, BLLT20, CL21]). That is, the observation that models with near-zero training error can achieve excellent test set performance. This insight was largely motivated by large deep learning architectures where this is commonplace [MBB18]. While this line of research does provide an important update to historic intuitions which linked perfect train set performance with poor generalization, it is entirely compatible with the ideas presented in this work. In particular, in Figure 20 we demonstrate that *within* the interpolation regime the number of effective parameters used on test examples and the predictive performance on those test examples can vary greatly. Therefore, benign overfitting may indeed result in excellent performance *without* contradicting the traditional notion of a U-shaped complexity curve. Additionally, note that while the double descent phenomenon is often presented as a consequence of benign overfitting in the interpolation regime, we show in Appendix E.2 that the location of the second descent is not

inherently linked to the interpolation regime at all in the non-deep experiments of [BHMM19] – but rather a consequence of the point of change between multiple mechanisms of increasing parameter count.

**Other related works.** Finally, we briefly discuss some other works that might provide further context on topics touched upon in this text. [BH89] study *linear* neural networks through the lens of principal component analysis. Recent theoretical work suggests that, when optimized using stochastic gradient descent, these networks tend to converge to low norm or rank solutions (e.g. [GWB$^+$17, ACHL19]). Aspects of double descent, which were referred to as *peaking* in older literature, have been associated with the eigenvalues of the sample covariance matrix in a limited setting involving linear classifiers as far back as [RD98]. Much of this work focused on increasing the number of samples for a fixed model (e.g. [Dui95, SD96]). Prior to being rediscovered in the context of neural networks and larger models, more recent work investigated the peaking phenomenon in modern machine learning settings such as semi-supervised learning [KL16].

# B    More on linear regression

In this section, we first present a brief refresher on SVD, PCA and PCR to provide some further background to the discussion in the main text and the following subsections. We then present a proof of Proposition 1 in Appendix B.2 and finally empirically analyze the effects of excess features on the conditioning of the PCR feature matrix in Appendix B.3.

## B.1    Background on SVD, PCA and PCR

In this section we provide a brief refresher on some of the mathematical tools that are fundamental to the insights presented in the main text. We begin with the singular value decomposition (SVD) which acts as a generalization of the standard eigendecomposition for a non-square matrix $\mathbf{X} \in \mathbb{R}^{n \times d}$ (see e.g. [Kal96, DFO20]). The SVD consists of the factorization $\mathbf{X} = \mathbf{U\Sigma V}^T$ where the columns of $\mathbf{U} \in \mathbb{R}^{n \times n}$ and rows of $\mathbf{V}^T \in \mathbb{R}^{d \times d}$ consist of the left and right *singular vectors* of $\mathbf{X}$ respectively and both form orthonormal bases of $\mathbf{X}$. Then, $\mathbf{\Sigma} \in \mathbb{R}^{n \times d}$ contains the so-called *singular values* $\sigma_1 \geq \ldots \geq \sigma_{\min(n,d)}$ along the diagonal with zeros everywhere else. Analogous to eigenvalues and eigenvectors, the SVD satisfies $\mathbf{XV}_i = \sigma_i \mathbf{U}_i$ for all $i \in 1, \ldots, \min(n, d)$. We note that for any $\sigma_i = 0$, the corresponding $\mathbf{V}_i$ lies in the nullspace of $\mathbf{X}$ (while $\mathbf{U}_i$ lies in the nullspace of $\mathbf{X}^T$). The SVD results in the geometric intuition of factorizing the transformation $\mathbf{X}$ into three parts consisting of a basis change in $\mathbb{R}^d$, followed by a scaling by the singular values that adds or removes dimensions, and then a further basis change in $\mathbb{R}^n$. We note that any columns of $\mathbf{U}$ and $\mathbf{V}$ corresponding to either null singular values (i.e. $\sigma_i = 0$) or the $\max(n, d) - \min(n, d)$ non-square dimensions of $\mathbf{X}$ are redundant in the SVD. Therefore we can equivalently express the SVD in its so-called *compact* form with these vectors and their corresponding rows/columns in $\mathbf{\Sigma}$ removed.

The SVD is intimately linked to dimensionality reduction through principal component analysis (PCA) [Pea01, FM23, WRR03]. Recall that PCA is a technique that finds a basis of a centered matrix $\mathbf{X}$ by greedily selecting each successive basis dimension to maximize the variance within the data. Typically, this is presented as selecting each basis vector $\mathbf{b}_i = \arg\max_{||\mathbf{b}||=1} ||\hat{\mathbf{X}}_i \mathbf{b}||^2$ where $\hat{\mathbf{X}}_i$ denotes the original (centered) data matrix $\mathbf{X}$ with the first $i - 1$ directions of variation removed (i.e. $\hat{\mathbf{X}}_i = \mathbf{X} - \sum_{k=1}^{i-1} \mathbf{b}_k \mathbf{b}_k^T \mathbf{X}$). By expanding $||\hat{\mathbf{X}}_i \mathbf{b}||^2 = \mathbf{b}^T \hat{\mathbf{X}}_i^T \hat{\mathbf{X}} \mathbf{b}^T$ we can notice that the basis vectors obtained are also the eigenvectors of $\mathbf{X}^T \mathbf{X}$ which is proportional to the sample covariance matrix. The dimensionality reduction then occurs by selecting a subset of these new coordinates, ordered by highest variance, to represent a low-rank approximation of $\mathbf{X}$. The relationship to the SVD is illuminated by noticing that, for a centered $\mathbf{X}$, we can use the singular value decomposition such that

$$
\begin{aligned}
\mathbf{X}^T \mathbf{X} &= (\mathbf{U\Sigma V}^T)^T (\mathbf{U\Sigma V}^T) \\
&= \mathbf{V\Sigma}^T \mathbf{\Sigma V}^T \\
&= \mathbf{V\Sigma}^2 \mathbf{V}^T.
\end{aligned}
$$

This reveals that $\mathbf{V}$, the orthonormal basis containing right singular vectors of $\mathbf{X}$ and obtained through the SVD, produces exactly the eigenvectors of $\mathbf{X}^T \mathbf{X}$ required for PCA. Then the PCA operation $PCA_k(\mathbf{X}) : \mathbb{R}^{n \times d} \to \mathbb{R}^{n \times k}$ (where $k \leq d$ is the user-defined number of principal components

retained) may be obtained as $PCA_k(\mathbf{X}) = \mathbf{X}\mathbf{V}_k = \mathbf{U}_k\mathbf{\Sigma}_k$ where we overload notation such that the subscript of each matrix denotes keeping only the vectors corresponding to those first $k$ principal components (ordered according to the magnitude of their singular values).

A natural idea that emerges from the description of PCA is to apply it in the supervised setting. Specifically, we might wish to first perform the unsupervised dimensionality reduction step of PCA followed by least squares linear regression on the transformed data. This is exactly the approach taken in principal component regression (PCR) [Ken57, Hot57]. Mathematically this can be expressed as solving $\hat{\boldsymbol{\beta}} = \min_{\boldsymbol{\beta}}(\mathbf{y} - \mathbf{X}\mathbf{V}_k\boldsymbol{\beta})^2$ for coefficients $\boldsymbol{\beta} \in \mathbb{R}^k$ and targets $\mathbf{y} \in \mathbb{R}^n$. Then the least squares solution takes the form $\hat{\boldsymbol{\beta}} = (\mathbf{V}_k^T\mathbf{X}^T\mathbf{X}\mathbf{V}_k)^{-1}\mathbf{V}_k^T\mathbf{X}^T\mathbf{y}$. The assumption of PCR is that the high variance components are most important for modeling the targets $\mathbf{y}$. While this is not necessarily always the case[7] [Jol82], PCR has historically been an important method in the statistician's toolbox (e.g. [HTF09]). In the main text we make use of the connection between PCR and min-norm least squares on underdetermined problems (i.e. a fat data matrix). In particular, we show that applying min-norm least squares in this setting is equivalent to applying PCR with all principal components retained. We provide a proof for this proposition in Appendix B.2. Of course, related mathematical connections between min-norm solutions and the singular value decomposition have been made in previous work (see e.g. [GVL13, Ch. 5.7]). The regularization effect of PCR also has a known connection to ridge regression where both methods penalize the coefficients of the principal components [HTF09, Ch. 3.5.1]. The former discards components corresponding to the smallest singular values while the latter shrinks coefficients in proportion to the size of the singular values[8].

## B.2    Proof of Proposition 1

**Proposition 1.** *[Min-norm least squares as dimensionality reduction.]  For a full rank matrix $\mathbf{X} \in \mathbb{R}^{n \times d}$ with $n < d$ and a vector of targets $\mathbf{y} \in \mathbb{R}^n$, the min-norm least squares solution $\hat{\boldsymbol{\beta}}^{MN} = \{\min_{\boldsymbol{\beta}} ||\boldsymbol{\beta}||_2^2 \colon \mathbf{X}\boldsymbol{\beta} = \mathbf{y}\}$ and the least squares solution $\hat{\boldsymbol{\beta}}^{SVD} = \{\boldsymbol{\beta} \colon \mathbf{B}\boldsymbol{\beta} = \mathbf{y}\}$ using the matrix of basis vectors $\mathbf{B} \in \mathbb{R}^{n \times n}$, constructed using the first $n$ right singular vectors of X, are equivalent; i.e. $\mathbf{x}^T\hat{\boldsymbol{\beta}}^{MN} = \mathbf{b}^T\hat{\boldsymbol{\beta}}^{SVD}$ for all $\mathbf{x} \in \mathbb{R}^d$ and corresponding basis representation $\mathbf{b} \equiv \mathbf{b}(\mathbf{x})$.*

*Proof.* We will use the compact singular value decomposition $\mathbf{X} = \mathbf{U}\mathbf{\Sigma}\mathbf{V}^T$ (i.e. with the $d - n$ redundant column vectors dropped from the full $\mathbf{\Sigma}$ and $\mathbf{V}$). Specifically, the columns of $\mathbf{U} \in \mathbb{R}^{n \times n}$ and $\mathbf{V}^T \in \mathbb{R}^{n \times d}$ form orthonormal bases and $\mathbf{\Sigma} \in \mathbb{R}^{n \times n}$ is a diagonal matrix with the singular values along the diagonal. The transformation of $\mathbf{X}$ to $\mathbf{B}$ is then given by $\mathbf{X}\mathbf{V} = \mathbf{U}\mathbf{\Sigma} = \mathbf{B}$. It is sufficient to show that for some test vector $\mathbf{x} \in \mathbb{R}^d$ in the original data space and its projection to the reduced basis space $\mathbf{b} = \mathbf{V}^T\mathbf{x} \in \mathbb{R}^n$, then the predictions $\hat{y}$ of both models are equal such that $\mathbf{x}^T\hat{\boldsymbol{\beta}}^{\text{MN}} = \mathbf{b}^T\hat{\boldsymbol{\beta}}^{\text{SVD}}$.

$$
\begin{aligned}
\hat{y}^{\text{MN}} = \mathbf{x}^T\hat{\boldsymbol{\beta}}^{\text{MN}} &= \mathbf{x}^T\mathbf{X}^T(\mathbf{X}\mathbf{X}^T)^{-1}\mathbf{y} \\
&= \mathbf{x}^T(\mathbf{U}\mathbf{\Sigma}\mathbf{V}^T)^T((\mathbf{U}\mathbf{\Sigma}\mathbf{V}^T)(\mathbf{U}\mathbf{\Sigma}\mathbf{V}^T)^T)^{-1}\mathbf{y} \\
&= \mathbf{x}^T\mathbf{V}\mathbf{\Sigma}\mathbf{U}^T(\mathbf{U}\mathbf{\Sigma}\mathbf{\Sigma}\mathbf{U}^T)^{-1}\mathbf{y} \\
&= \mathbf{x}^T\mathbf{V}\mathbf{B}^T(\mathbf{B}\mathbf{B}^T)^{-1}\mathbf{y} \\
&= \mathbf{b}^T\mathbf{B}^T(\mathbf{B}\mathbf{B}^T)^{-1}\mathbf{y} \\
&= \mathbf{b}^T\hat{\boldsymbol{\beta}}^{\text{SVD}} \\
&= \hat{y}^{\text{SVD}}
\end{aligned}
$$

$\square$

---

[7]Indeed, an alternative approach is to use standard variable selection techniques on the constructed dimensions of the transformed data (see e.g. [SKL92]).

[8]This connection explains why the explicit regularization with the ridge penalty in [HMRT22] and the implicit regularization of min-norm solutions have a consistent shrinking effect on test loss.

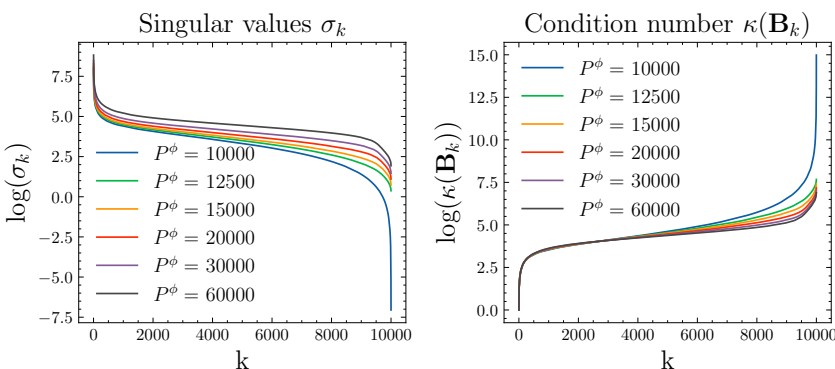

Figure 10: **The effect of $P^\phi$ on the conditioning of the PCR feature matrix.** Singular values $\sigma_k$ (left) and $\kappa(\mathbf{B}_k)$ (right) derived from the random Fourier feature matrix $\mathbf{\Phi}$, as a function of $k$ and for different raw number of features $P^\phi$.

### B.3 Understanding the impact of $P^{ex}$: Examining its impact on singular values of the PCR feature matrix

In this section, we empirically investigate *how* increasing $P^{ex}$ for a fixed number of $P^{PC}$ results in a gradual improvement in the quality of a basis of fixed dimension, revisiting the experimental setting of Sec. 3.2 in the main text.

In particular, we consider how excess features affect the conditioning of the feature matrix $\mathbf{B}_k = PCA_k(\mathbf{X}) = \mathbf{X}\mathbf{V}_k = \mathbf{U}_k\mathbf{\Sigma}_k$, used in principal components regression, for different values of $k \equiv P^{PC}$. As noted in [PKB19], the condition number $\kappa(\mathbf{A})$ of a matrix $\mathbf{A}$ measures how much errors in $\mathbf{y}$ impact solutions when solving a system of equations $\mathbf{A}\beta = \mathbf{y}$, and is given by $\kappa(\mathbf{A}) = \frac{\sigma_{\max}(\mathbf{A})}{\sigma_{\min}(\mathbf{A})}$, the ratio of maximal to minimal singular value of $\mathbf{A}$. Because in principal components regression $\mathbf{B}_k$ is constructed from the top $k \in \{1, \ldots, n\}$ principal components, it is easy to see that $\sigma_{\max}(\mathbf{B}_k) = \sigma_1$ and $\sigma_{min}(\mathbf{B}_k) = \sigma_k$ – the largest and k-th largest singular value of $\mathbf{X}$ respectively (as defined in Appendix B.1) – so that in principal components regression we generally have $\kappa(\mathbf{B}_k) = \frac{\sigma_1}{\sigma_k}$.

To understand how the raw number of features $P^\phi$ in the experimental setup of Sec. 3.2 in the main text impacts basis quality, it is thus instructive to study the behavior of the singular values of the random Fourier feature matrix $\mathbf{\Phi}$ as $P^\phi$ grows. In Fig. 10, we therefore plot $\sigma_k$ and $\kappa(\mathbf{B}_k)$ for different values of $P^\phi$ and observe that indeed, as $P^\phi$ (and thereby $P^{ex} = P^\phi - k$) increases, $\sigma_k$ substantially increases in magnitude especially for large $k \equiv P^{PC}$. This provides empirical evidence that the conditioning of feature matrices for principal components regression at large values of $k$ indeed substantially improves as we add excess raw features (i.e. $\kappa(\mathbf{B}_k)$ decreases substantially – particularly at large $k$ – as can be seen in the right panel of Fig. 10).

## C  More on smoothers: Bias-variance, on the impossibility of in-sample double descent and derivation of smoothing matrices

In this section, we give some further background on the well-known relationship between bias and variance in smoothers, then discuss the impossibility of double descent in prediction error in *fixed design* settings, and finally derive the smoothing matrices implied by the ML methods used in the main text.

### C.1  Background: Bias and variance in smoothers

In this section, we discuss the well-known bias-variance relationship in smoothers (which indeed determines the U-shape of their original complexity-error curves), following the discussion in [HT90, Sec. 3.3]. Under the standard assumption that targets are generated with true expectation $f^*(x_0)$ and homoskedastic variance $\sigma^2$, the bias and variance of a smoother $\hat{\mathbf{s}}(\cdot)$ evaluated at test point $x_0$ (for

*fixed* input points $\{x_i\}_{i \in \mathcal{I}_{\text{train}}}$ and *fixed* smoother $\hat{s}(\cdot)$), can be written as:

$$\text{Bias}(\hat{f}, x_0) = f^*(x_0) - \sum_{i \in \mathcal{I}_{\text{train}}} \hat{s}^i(x_0) f^*(x_i) \tag{3}$$

$$\text{Var}(\hat{f}, x_0) = \text{Var}(\hat{s}(x_0) \mathbf{y}_{\text{train}}) = ||\hat{s}(x_0)||^2 \sigma^2 \tag{4}$$

Note that some of the historical intuition behind the U-shaped bias-variance tradeoff with model complexity on the x-axis appears to lead back to exactly the comparison between these two terms evaluated for *training inputs* in smoothers –[Nea19] trace explicit discussion of bias-variance tradeoff back to at least [HT90] (who indeed discuss it in the context of smoothers), preceding [GBD92]'s influential machine learning paper on the topic. A bias-variance tradeoff is easily illustrated by considering e.g. a kNN smoother[9]: a 1NN estimator has *no bias* when predicting outcomes for a training input but variance $\sigma^2$, while a $k(>1)$NN estimator generally incurs some bias but has variance $\sigma^2/k$ – we thus expect a U-shaped curve in squared prediction error as we vary $k$. Similarly, note that for linear regression on a $p < n$-dimensional subset of the input, the bias of the fit generally decreases in $p$, while the variance increases in $p$ as $\frac{1}{n} \sum_{i=1}^{n} Var(\hat{s}(x_i)) = \frac{p\sigma^2}{n}$ [HTF09] – leading to another well-known example of a U-shaped generalization curve (in $p < n$).

When the expected mean-squared error (MSE) is the loss function of interest, a bias-variance tradeoff can also be motivated more explicitly by noting that the MSE can be decomposed as

$$\mathbb{E}[(Y - \hat{f}(x_0))^2 | X = x_0] = \sigma^2 + \text{Bias}^2(\hat{f}, x_0) + \text{Var}(\hat{f}, x_0) \tag{5}$$

### C.2 On the impossibility of a second prediction error descent in the interpolation regime in *fixed design* settings

[BHMM19] provide a discussion of possible cultural and practical reasons for the historical absence of double descent shapes in the statistics literature. They note that observing double descent in the number of parameters requires parametric families in which complexity can arbitrarily be controlled and that therefore the classical statistics literature – which either focuses either on linear regression in fixed $p < n$ settings or considers highly flexible nonparametric classes but then applies heavy regularization – could not have observed it because of this cultural focus on incompatible model classes (in which [BHMM19] argue double descent cannot emerge due to a lack of interpolation). Through our investigation, we find another fundamental reason for the historical absence of double descent shapes in statistics: even when using a sufficiently flexible parametric class (e.g. unregularized RFF linear regression), a second descent in the interpolation regime *cannot* emerge in fixed design settings, the primary setting considered in the early statistics literature – where much of the intuition regarding the bias-variance tradeoff was initially established.

After recalling some definitions in Appendix C.2.1, we demonstrate this impossibility of a second descent in fixed design settings for general interpolating predictors in Appendix C.2.2 below, and make some additional observations specific to smoothers in Appendix C.2.3.

#### C.2.1 Preliminaries.

We begin by recalling the definition of the fixed design setting of statistical learning.

**Definition 2** (Fixed design setting). *In the fixed design setting, the inputs $x_1, \ldots, x_n$ are deterministic, while their associated targets are random (e.g. $y_i = f(x_i) + \epsilon_i$ where $\epsilon_i$ is a random noise term).*

The fixed design setting was a core consideration of much of the statistical literature [RT19] and was suitable for several traditional applications in e.g. agriculture. While many modern applications require a *random design setting* in which the inputs are also random, several modern applications are still suitably modeled within this framework (e.g. image denoising). Next, we will define in-sample prediction error in line with [HTF09].

**Definition 3** (In-sample prediction error). *Given a set of fixed design points $x_1, \ldots, x_n$, training targets $y_1, \ldots, y_n$ and some model $f$, suppose we observe a new set of random responses at each of*

---

[9]Recall that, possibly counterintuitively, a kNN smoother is most complex when $k = 1$, where at every training input the prediction is simply the original training label, and least complex when $k = n$, where it just outputs a sample average

the training points denoted $y_i^0$. Then the expected in-sample prediction error for some loss function $\mathcal{L}$ is given by

$$ERR_{in} = \frac{1}{n} \sum_{i=1}^{n} \mathbb{E}_{y^0}[\mathcal{L}(y_i^0, f(x_i))|\{(x_1, y_1), \ldots, (x_n, y_n)\}]$$

*Remark:* Note that in the fixed design setting, the expected in-sample prediction error is indeed the natural quantity of interest at test time.

Finally, we define an interpolating predictor below.

**Definition 4** (Interpolating predictor.). *A predictor $\hat{f}(\cdot)$ interpolates the training data if for $y_i^{train} \in \mathbb{R}, \forall i \in \mathcal{I}_{train}$*

$$\hat{f}(x_i) = y_i^{train} \tag{6}$$

### C.2.2 Main result: General impossibility of a second descent in test error for the interpolation regime in fixed design settings

We may now proceed to the main observation of this section. That is, in the fixed design setting a second descent in test error observed in the interpolation regime *cannot* occur. Intuitively, this is because any improvement in fixed-design test-time performance by a more complex model would require sacrificing the perfect performance on the train set. As a consequence, while – as we observe in the main text – different interpolating predictors can have *wildly different (out-of-sample) generalization performance*, we show below that all models in the interpolation regime *have identical expected in-sample prediction error* $ERR_{in}$. That is, increasing the raw number of parameters $p$ past $n$, e.g. in min-norm linear regression, does not and can not lead to a second descent in $ERR_{in}$. We express this more formally below.

**Proposition 2** (Impossibility of a second error descent in the interpolation regime for fixed design settings.). *Suppose we observe inputs $x_1, \ldots, x_n$ in a fixed design prediction problem, with associated random labels $y_1^{train}, \ldots, y_n^{train}$ in the training set. For all interpolating models $f$, test-time predictions will be identical and, therefore, a second descent in $ERR_{in}$ in this regime is impossible.*

*Proof.* By definition of fixed design problems, our test set consists of examples in which the features are duplicates from the training set and the targets are new realizations from some target distribution. Then the test dataset consists of the pairs $(x_1, y_1^{\text{test}}), \ldots, (x_n, y_n^{\text{test}})$. For any interpolating model $f$ we can calculate the test loss as $\mathcal{L}^{\text{test}} = \sum_{i=1}^{n} \mathcal{L}(f(x_i), y_i^{\text{test}})$. But by Definition 4 (and because interpolating models achieve zero training error), we know that $f(x_i) = y_i^{train}$. Thus $\mathcal{L}^{\text{test}} = \sum_{i=1}^{n} \mathcal{L}(y_i^{train}, y_i^{\text{test}})$, and since this is not a function of $f$, this quantity is independent of the number of parameters (or any other property) of $f$. Therefore, any change in $\mathcal{L}^{\text{test}}$ (or expectations thereof) in the interpolation regime is impossible - including the second descent necessary for creating a double descent shape. $\square$

While most, if not all, discussion of the double descent phenomenon in the ML literature is not within the fixed design setting, we believe that the impossibility of double descent in this setting provides a new useful partial explanation for its historical absence from the statistical discourse.

### C.2.3 Additional observations specific to smoothers

For the special case of interpolating linear smoothers, we make an additional interesting observation with regard to their smoothing weights and implied effective number of parameters. To do so, first recall that a linear smoother is defined as follows:

**Definition 5** (Linear smoother). *A smoother with weights $\hat{\mathbf{s}}(x_0) \equiv \hat{\mathbf{s}}(x_0; \mathbf{X}_{train}, \mathbf{y}_{train})$ is linear if $\hat{\mathbf{s}}(x_0)$ does not depend on $\mathbf{y}_{train}$; that is $\hat{\mathbf{s}}(x_0; \mathbf{X}_{train}, \mathbf{y}^1) = \hat{\mathbf{s}}(x_0; \mathbf{X}_{train}, \mathbf{y}^2)$ for $\forall \mathbf{y}^1, \mathbf{y}^2 \in \mathbb{R}^n$*

We are now ready to show in Prop. 3 that *all* interpolating linear smoothers must have the same smoother weights and effective parameters for training inputs.

**Proposition 3** (Interpolating linear smoother.). *Any interpolating linear smoother must have identical smoother weights $\hat{\mathbf{s}}(x_i) = \mathbf{e}_i$ for all $i \in \mathcal{I}_{train}$. This immediately implies also that all interpolating linear smoothers use the same effective number of parameters $p_e = n$ in fixed design settings.*

*Proof.* In linear smoothers, $\hat{\mathbf{s}}(x_i)$ can not depend on $\mathbf{y}_{\text{train}}$ by definition. To have $\hat{\mathbf{s}}(x_i)\mathbf{y}'_{\text{train}} = y'_i$ for all $i \in \mathcal{I}_{\text{train}}$ and *any possible* training outcome realization $\forall \mathbf{y}'_{\text{train}} \in \mathbb{R}^n$, we must have $\hat{\mathbf{s}}(x_i) = \mathbf{e}_i$ for $i \in \mathcal{I}_{\text{train}}$.

From this, as $||\mathbf{e}_i||^2 = 1$, it also follows that $p_{\hat{\mathbf{s}}}^{\text{train}} = p_e = \sum_{i=1}^n ||\mathbf{e}_i||^2 = n$. In fixed design settings, training and testing inputs are identical so $p_{\hat{\mathbf{s}}}^{\text{test}} = n$. $\square$

Note that this specific property does not hold for more general (non-linear) interpolating smoothers: whenever $\mathbf{y}_{train}$ contains duplicates, a non-linear interpolating smoother *can* have $\hat{\mathbf{s}}(x_i) \neq \mathbf{e}_i$ for some $i \in \mathcal{I}_{\text{train}}$. When all $y_i$ observed at train-time are unique, then it is easy to see that even non-linear interpolating smoothers must have $\hat{\mathbf{s}}(x_i) = \mathbf{e}_i$ for all $i \in \mathcal{I}_{\text{train}}$.

### C.3 Derivation of smoothing weights of linear regression, trees and boosting

#### C.3.1 Smoothing weights for RFF linear regression

Derivation of the smoothing weights for the RFF linear regression example is trivial, as linear regression is a textbook example of a smoother [HT90]. To see this, let $\phi(x_0) = (\phi_1(x_0), \ldots, \phi_P(x_0))$ denote the $P^\phi \times 1$ row-vector of RFF-projections of *arbitrary* admissible $x_0 \in \mathbb{R}^d$ and let $\boldsymbol{\Phi}_{\text{train}} = [\phi(x_1)^T, \ldots, \phi(x_n)^T]^T$ denote the $n \times P^\phi$ design matrix of the training set and note that the least squares prediction obtained through the min-norm solution can be written as

$$\hat{f}^{LR}(x_0) = \phi(x_0)\beta_{LR} = \hat{s}^{LR}(x_0)\mathbf{y}_{\text{train}} = \begin{cases} \phi(x_0)(\boldsymbol{\Phi}_{\text{train}}^T\boldsymbol{\Phi}_{\text{train}})^{-1}\boldsymbol{\Phi}_{\text{train}}^T\mathbf{y}_{\text{train}} & \text{if } P^\phi < n \\ \phi(x_0)\boldsymbol{\Phi}_{\text{train}}^T(\boldsymbol{\Phi}_{\text{train}}\boldsymbol{\Phi}_{\text{train}}^T)^{-1}\mathbf{y}_{\text{train}} & \text{if } P^\phi \geq n \end{cases} \tag{7}$$

i.e.

$$\hat{\mathbf{s}}^{LR}(x_0) = \begin{cases} \phi(x_0)(\boldsymbol{\Phi}_{\text{train}}^T\boldsymbol{\Phi}_{\text{train}})^{-1}\boldsymbol{\Phi}_{\text{train}}^T & \text{if } P^\phi < n \\ \phi(x_0)\boldsymbol{\Phi}_{\text{train}}^T(\boldsymbol{\Phi}_{\text{train}}\boldsymbol{\Phi}_{\text{train}}^T)^{-1} & \text{if } P^\phi \geq n \end{cases} \tag{8}$$

Note also that by design, as alluded to in Appendix C.2.3, when $P^\phi \geq n$ we have $\boldsymbol{\Phi}_{\text{train}}\boldsymbol{\Phi}_{\text{train}}^T(\boldsymbol{\Phi}_{\text{train}}\boldsymbol{\Phi}_{\text{train}}^T)^{-1} = \mathbf{I}_n$, implying that that $\hat{s}^{LR}(x_i) = \mathbf{e}_i$ for any $x_i$ observed at train time.

#### C.3.2 Smoothing weights for trees and forests

Similarly, we can exploit that trees are sometimes considered *adaptive nearest neighbor methods* [HTF09, Ch. 15.4.3], i.e. nearest neighbor estimators with an adaptively constructed (outcome-oriented) kernel (or distance measure). To see this, note that regression trees, that is trees trained to minimize the squared loss and/or classification trees that make predictions through averaging (not voting), can all be understood as issuing predictions that simply average across training instances within a terminal leaf. To make this more formal, consider that any tree with $P^{leaf}$ leaves can be represented by $P^{leaf}$ contiguous axis-aligned hypercubes $l_p$, which store the mean outcome of all training examples that fall into this leaf: If we denote by $l(x_0)$ the leaf that $x_0$ falls into and $n_{l(x_0)} = \sum_{i=1}^n \mathbf{1}\{l(x_0) = l(x_i)\}$ the number of training examples in this leaf, then we have that

$$\hat{f}(x_0) = \frac{1}{n_{l(x_0)}} \sum_{i=1}^n \mathbf{1}\{l(x_0) = l(x_i)\}y_i \tag{9}$$

giving

$$\hat{\mathbf{s}}^{tree}(x_0) = \left( \frac{\mathbf{1}\{l(x_0) = l(x_1)\}}{n_{l(x_0)}}, \ldots, \frac{\mathbf{1}\{l(x_0) = l(x_1)\}}{n_{l(x_0)}} \right) \tag{10}$$

Further, we note that sums $\hat{f}_\Sigma(x_0) = \sum_{m=1}^M \hat{f}^m(x_0)$ of smoothers $\hat{f}^m(x_0) = \hat{s}^m(x_0)\mathbf{y}_{\text{train}}$, $m = 1, \ldots, M$ are also smoothers with $\hat{s}^\Sigma(x_0) = \sum_{m=1}^M \hat{s}^m(x_0)$. This immediately implies that the tree ensembles considered in Sec. 2.1 are also smoothers, and, as we show in Appendix C.3.3 below, so are boosted trees (and ensembles thereof) as considered in Sec. 2.2.

**Remark:** We note that, although the smooth $\hat{s}^{tree}(x_0)$ can be *written* in linear form without explicit reference to $\mathbf{y}_{\text{train}}$, trees (and their sums) are not truly linear smoothers because the smoother $\hat{s}^{tree}(\cdot)$ does depend in its construction on $\mathbf{y}_{\text{train}}$, reflecting their epithet as *adaptive smoothers* [HTF09, Ch.

15.4.3]. This mainly implies that when calculating bias and variance, we can no longer consider $\hat{s}(x_0)$ as fixed and Eqs. (3) and (4) no longer hold exactly. However, this is not an issue for our purposes as we are not interested in actually estimating bias and variance in this work, rather we are interested in smoothers because – as discussed in Sec. 4 and Appendix D – they allow us to compute the most useful measure of effective parameters. Conceptually, this allows us to assess the *effective amount* of smoothing used, which can also be interpreted relative to e.g. a standard kNN estimator. Because for kNN estimators we have $p_{\hat{s}}^0 = \frac{n}{k}$ and for trees and forests it holds that $\sum_{i=1}^n \hat{s}^i(x_0) = 1$, the quantity $\tilde{k}_{\hat{s}}^0 = \frac{n}{p_{\hat{s}}^0}$ admits an interesting interpretation as measuring the *effective number of nearest neighbors* examples have in a tree or forest (and this interpretation does *not* require fixed $\hat{s}$).

### C.3.3 Smoothing weights for boosting

Deriving the smoothing weights for gradient boosting is more involved. To do so, we first re-state the gradient boosting algorithm (as presented in [HTF09, Ch. 10.10]) and then recursively construct weights from it.

**The gradient boosted trees algorithm.** We consider gradient boosting with learning rate $\eta$ and the squared loss as $L(\cdot, \cdot)$:

1. Initialize $f_0(x) = 0$
2. For $p \in \{1, \ldots, P^{boost}\}$
   (a) For $i \in \{1, \ldots, n\}$ compute
   $$g_{i,p} = -\left[\frac{\partial L(y_i, f(x_i))}{\partial f(x_i)}\right]_{f=f_{p-1}} \tag{11}$$
   (b) Fit a regression tree to $\{(x_i, g_{i,p})\}_{i=1}^n$, giving leaves $l_{jp}$ for $j = 1, \ldots, J_p$
   (c) Compute optimal predictions for each leaf $j \in \{1, \ldots, J_p\}$:
   $$\gamma_{jp} = \arg\min_{\gamma \in \mathbb{R}} \sum_{x_i \in l_{jp}} L(y_i, f_{p-1}(x_i) + \gamma) = \frac{1}{n_{l_{jp}}} \sum_{x_i \in l_{jp}} (y_i - f_{p-1}(x_i)) \tag{12}$$
   (d) Denote by $\tilde{f}_p(x) = \sum_{j=1}^{J_p} \mathbf{1}\{x \in l_{jp}\}\gamma_{jp}$ the predictions of the tree built in this fashion
   (e) Set $f_p(x) = f_{p-1}(x) + \eta\tilde{f}_p(x)$
3. Output $f(x) = f_{P^{boost}}(x)$

**Recursive construction of smoothing weights.** Equation (12) highlights that we can *recursively* construct smoothing weights similarly as for trees. In fact, note that in the first boosting round, $\tilde{f}_1(x)$ is a standard tree with smoothing weights $\hat{s}^{tree,\tilde{f}_1}(\cdot)$ computed as in Eq. (10), so that $f_1(x) = \eta\tilde{f}_1(x)$ has smoothing weights $\hat{s}^{boost,1}(\cdot) = \eta\hat{s}^{tree,\tilde{f}_1}(\cdot)$. Now we can recursively consider the smoothing weights associated with newly added residual trees $\tilde{f}_p(x)$ by realizing that its predictions can be written as a function of standard tree weights $\hat{s}^{tree,\tilde{f}_p}(\cdot)$ and a correction depending on $\hat{s}^{boost,1}(\cdot)$ alone :

$$\tilde{f}_p(x) = \sum_{j=1}^{J_p} \mathbf{1}\{x \in l_{jp}\}\gamma_{jp} = \sum_{j=1}^{J_p} \mathbf{1}\{x \in l_{jp}\} \left(\frac{1}{n_{l_{jp}}} \sum_{x_i \in l_{jp}} (y_i - f_{p-1}(x_i))\right)$$

$$= \underbrace{\sum_{j=1}^{J_p} \mathbf{1}\{x \in l_{jp}\} \left(\frac{1}{n_{l_{jp}}} \sum_{x_i \in l_{jp}} y_i\right)}_{\text{Standard tree prediction: } \hat{s}^{tree,\tilde{f}_p}(x)\mathbf{y}_{train}} - \underbrace{\sum_{j=1}^{J_p} \mathbf{1}\{x \in l_{jp}\} \left(\frac{1}{n_{l_{jp}}} \sum_{x_i \in l_{jp}} \hat{s}^{boost,p-1}(x_i)\mathbf{y}_{train}\right)}_{\text{Residual correction: } \hat{s}^{corr,f_p}(x)\mathbf{y}_{train}} \tag{13}$$

$$= (\hat{s}^{tree,\tilde{f}_p}(x) - \hat{s}^{corr,f_{p-1}}(x))\mathbf{y}_{train} \tag{14}$$

where, using the $1 \times J_p$ indicator vector $\mathbf{e}_{l^p(x)} = (\mathbf{1}\{x \in l_1\}, \ldots, \mathbf{1}\{x \in l_{J_p}\})$, we have

$$\hat{s}^{corr,f_p}(x) = \mathbf{e}_{l^p(x)}\hat{\mathbf{R}}^p \tag{15}$$

with $\hat{\mathbf{R}}^p$ the $J_p \times n$ leaf-residual correction matrix with $j-$th row given by

$$\hat{\mathbf{R}}^p = \frac{1}{n_{l_{jp}}} \sum_{x_i \in l_{jp}} \hat{\mathbf{s}}^{boost,p-1}(x_i) \tag{16}$$

Thus, we have constructed the smoothing matrix for gradient boosting recursively as:

$$\hat{\mathbf{s}}^{boost,p}(\cdot) = \hat{\mathbf{s}}^{boost,p-1}(\cdot) + \eta \left( \hat{\mathbf{s}}^{tree,\tilde{f}_p}(\cdot) - \hat{\mathbf{s}}^{corr,f_p}(\cdot) \right) \tag{17}$$

## D  More on effective parameters

In what follows, we discuss the different effective parameter definitions considered in the smoothing literature, and follow closely the presentation in [HT90, Ch.3.4 & 3.5] and [HTF09, Ch. 7.5]. Recall that, as discussed in Sec. 4 of the main text, these definitions are all naturally *calibrated* towards linear regression so that, as we might desire, effective and raw parameter numbers are equal in the case of ordinary linear regression with $p < n$.

To understand their motivation, we introduce some further notation. Assume that outcomes are generated as $y = f^*(x) + \epsilon$ with unknown expectation $f^*(x)$ and $\epsilon$ a zero-mean error term with variance $\sigma^2$. Let $\mathbf{f}^* = [f^*(x_1), \ldots, f^*(x_n)]^T$ denote the vector of true expectations. Further, let $\hat{\mathbf{S}} = [\hat{\mathbf{s}}(x_1)^T, \ldots, \hat{\mathbf{s}}(x_n)^T]^T$ denote the smoother matrix for all input points, so that we can write the vector of in-sample predictions as $\hat{\mathbf{f}} = \hat{\mathbf{S}}\mathbf{y}_{\text{train}}$. For linear smoothers (and/or fixed $\hat{\mathbf{S}}$), bias and variance at any input point are given by Eqs. (3) and (4), so that we have $\frac{1}{n} \sum_{i \in \mathcal{I}_{train}} bias^2(\hat{f}(x_i)) = \frac{1}{n} \mathbf{b}_{\hat{\mathbf{S}}}^T \mathbf{b}_{\hat{\mathbf{S}}}$ for bias vector $\mathbf{b}_{\hat{\mathbf{S}}} = (\mathbf{f}^* - \hat{\mathbf{S}}\mathbf{f}^*)$ and $\frac{1}{n} \sum_{i \in \mathcal{I}_{train}} var(\hat{f}(x_i)) = \frac{\sigma^2}{n} tr(\hat{\mathbf{S}}\hat{\mathbf{S}}^T)$.

**Covariance-based effective parameter definition.** In this setup, one way of defining the effective number of parameters that is very commonly used is to consider the covariance between predictions $\hat{f}(x_i)$ and the observed training label $y_i$ [HTF09]:

$$p_e^{cov} = \frac{\sum_{i=1}^n cov(y_i, \hat{f}(x_i))}{\sigma^2} = tr(\hat{\mathbf{S}}) \tag{18}$$

As expected, $tr(\hat{\mathbf{S}}) = p$ in ordinary linear regression ($p < n$).

**Error-based effective parameter definition.** Another way considers the residual sum of squares (RSS) in the training sample $RSS(\hat{\mathbf{f}}) = \sum_{i=1}^n (y_i - \hat{f}(x_i))^2 = (\mathbf{y}_{train} - \hat{\mathbf{S}}\mathbf{y}_{train})^T (\mathbf{y}_{train} - \hat{\mathbf{S}}\mathbf{y}_{train})$ and notes that it has expected value:

$$\mathbb{E}[RSS(\hat{\mathbf{f}})] = (n - tr(2\hat{\mathbf{S}} - \hat{\mathbf{S}}\hat{\mathbf{S}}^T))\sigma^2 + \mathbf{b}_{\hat{\mathbf{S}}}^T \mathbf{b}_{\hat{\mathbf{S}}} \tag{19}$$

and uses

$$n - p_e^{err} = n - tr(2\hat{\mathbf{S}} - \hat{\mathbf{S}}\hat{\mathbf{S}}^T) \tag{20}$$

because in the linear regression case the degrees of freedom for error are $n - p$ [HT90] and $tr(2\hat{\mathbf{S}} - \hat{\mathbf{S}}\hat{\mathbf{S}}^T) = p$.

**Variance-based effective parameter definition.** Finally, the variance-based definition $p_e$ discussed in the main text is motivated from $\sum_{i=1}^n var(\hat{f}(x_i)) = \sigma^2 \sum_{i=1}^n ||\hat{\mathbf{s}}(x_i)||^2 = \sigma^2 tr(\hat{\mathbf{S}}\hat{\mathbf{S}}^T)$ giving

$$p_e \equiv p_e^{var} = tr(\hat{\mathbf{S}}\hat{\mathbf{S}}^T) \tag{21}$$

as $\sum_{i=1}^n var(\hat{f}(x_i) = p\sigma^2$ in linear regression.

Note that, in ordinary linear regression with $p < n$ and full rank design matrix, $\hat{\mathbf{S}}$ is idempotent so that $tr(\hat{\mathbf{S}}\hat{\mathbf{S}}^T) = tr(\hat{\mathbf{S}}) = tr(2\hat{\mathbf{S}} - \hat{\mathbf{S}}\hat{\mathbf{S}}^T) = rank(\hat{\mathbf{S}}) = p$.

**Why did we choose $p_e^{var}$?** We chose to adapt $p_e^{var}$ because it is the definition that can most obviously be adapted to out-of-sample settings: both the error-based definition and the covariance-based definition focus on the effect a training outcome has in *predicting itself* (i.e. overfitting on the own label), while for out-of-sample prediction, this is not of interest. Instead, we therefore rely on the variance-based definition because $||\hat{\mathbf{s}}(x_0)||^2$ can not only be computed for every input $x_0$ but also

admits a meaningful interpretation as measuring the amount of smoothing over the different training examples: for an averaging smoother with $\sum_{i=1}^{n} \hat{\mathbf{s}}^i(x_0) = 1$, the simplest and smoothest prediction is a sample average with $\hat{\mathbf{s}}^i(x_0) = \frac{1}{n}$ – which minimizes $||\hat{\mathbf{s}}(x_0)||^2 = \sum_{i=1}^{n} \frac{1}{n^2} = \frac{1}{n}$. Conversely, the least smooth weights assign all weight to a single training observation $j$, i.e. $\hat{\mathbf{s}}(x_0) = \mathbf{e}_j$ so that $||\hat{\mathbf{s}}(x_0)||^2 = 1$ is maximized.

**Effective parameters for methods that cannot be written in linear form.** The effective parameter definitions considered above all have in common that they require a way of writing predictions $\hat{y}$ as an explicit function of $\mathbf{y}_{train}$ – which is possible for the three ML methods considered in this paper, but not generally the case. Instead, for loss-based methods like neural networks, [HTF09, Ch. 7.7] note that one can make a quadratic approximation to the error function $R(\mathbf{w})$ and get

$$p_e^\alpha = \sum_{j=1}^{p} \frac{\theta_j}{\theta_j + \alpha} \tag{22}$$

where $\theta_p$ are the eigenvalues of the Hessian $\partial^2 R(\mathbf{w})/\partial \mathbf{w} \partial \mathbf{w}^T$ and $\alpha$ is the weight-decay penalty used. This proxy effective parameters, motivated in a Bayesian context [Mac91], is used in [MBW20]'s study of effective parameters used in overparameterized neural networks. We note that, in addition to not being applicable to our tree-based experiments (which do not possess a differentiable loss function), this effective parameter proxy is also usually motivated in the context where *in-sample* prediction error is of interest [Moo91] – and might therefore also need updating to reflect the modern focus on generalization to unseen inputs. This is definitely the case in the context of linear regression: to make Eq. (22) applicable to the unregularized/ridgeless regression setup we consider, one needs to let $\alpha$ tend to 0. In this case, the Hessian is $\mathbf{X}'\mathbf{X}$ and we will have that $p_e^\alpha \to \sum_{j=1}^{p} \mathbf{1}\{\theta_j > 0\} = rank(\mathbf{X}'\mathbf{X}) = rank(\mathbf{X}) = min(n, p)$ – which will be constant at value $n$ once $p > n$, just like $p_{\hat{\mathbf{s}}}^{\text{train}}$.

**Other related concepts.** [DLM20] theoretically study implicit regularization of min-norm solutions by computing effective ridge penalties $\lambda_n$ implied by min-norm solutions by solving $n = tr(\Sigma_\mu(\Sigma_\mu + \lambda_n I_n)^{-1})$ with $\Sigma_\mu = E[XX^T]$. Further, building on the principle of minimum description length (MDL), [DSYW20] define a complexity measure in the context of linear regression via an optimality criterion over the encodings induced by a good Ridge estimator class.

# E  Additional results

In this section, we first provide further discussion of the experimental setup in Appendix E.1 and then present additional results: In Appendix E.2 we show that the location of the second descent can be arbitrarily controlled for all methods under consideration, in Appendix E.3 we plot binary test error, in Appendix E.4 we plot training error, in Appendix E.5 we provide anecdotal evidence that $p_{\hat{\mathbf{s}}}^{\text{test}}$ could be used for model selection and in Appendix E.6 we finally present results using further datasets.

## E.1  Experimental setup

Our experimental setup largely replicates that of [BHMM19]. We describe the datasets, computation, and each experiment in detail in this section. Code is provided at `https://github.com/alanjeffares/not-double-descent`.

**Datasets** - We perform our experiments in the main text on the MNIST image recognition dataset [LBBH98] with inputs of dimension 784. In this section, we also repeat these experiments on the SVHN digit recognition task [NWC+11] as well as the CIFAR-10 object classification tasks [KH+09], both containing inputs of dimension 1024 as color images have been converted to grayscale. All inputs are normalized to lie in the range $[0, 1]$. All three tasks are 10 class classification problems. Furthermore, we randomly sample a subset of 10,000 training examples and use the full test set in our evaluation. In SVHN we randomly sample a balanced test set of 10,000 examples. We note that the less common choice of using squared error for this classification problem in [BHMM19] is supported by the same authors' work in [HB21, MNS+21] on the utility of the squared loss for classification.

**Compute** - All experiments are performed on an Azure FX4mds. This machine runs on an Intel Xeon Gold 6246R (Cascade Lake) processor with 84 GiB of memory. The majority of the computational

cost was due to matrix inversions of a large data matrix in the case of linear regression experiments and fitting a large number of deep trees in the tree-based experiments. Each of the three experiments had a total runtime in the order of hours.

**Trees (Sec. 2.1)** - We use regression trees and random forests as implemented in scikit-learn [PVG$^+$11] for our tree experiments, which in turn implement an optimized version of the CART algorithm [BFSO84]. As in [BHMM19], we disable bootstrapping throughout and consider random feature subsets of size $\sqrt{d}$ at each split. Like [BHMM19] rely on 10 one-vs-all models due to the multi-class nature of the datasets.

**Gradient Boosting (Sec. 2.2)** - Gradient boosting is implemented as described in [HTF09, Ch. 10.10] and restated in Appendix C.3.3, using trees with $P^{leaf} = 10$ and a random feature subset of size $\sqrt{d}$ considered at each split, and learning rate $\gamma = .85$ as used in [BHMM19]. We rely on the scikit-learn [PVG$^+$11] implementation `GradientBoostingRegressor` to make use of the squared loss function, and like [BHMM19] rely on 10 one-vs-all models to capture the multi-class nature of the datasets. We create ensembles of $P^{ens}$ different boosted models by initialing them with different seeds.

**Linear regression (Sec. 3)** - As described in the main text we begin with a random Fourier Features (RFF) basis expansion. Specifically, given input $\mathbf{x} \in \mathbb{R}^d$, the number of raw model parameters $P^\phi$ is controlled by randomly generating features $\phi_p(\mathbf{x}) = \text{Re}(\exp^{\sqrt{-1}\mathbf{v}_p^T \mathbf{x}})$ for all $p \leq P^\phi$, where each $\mathbf{v}_p \overset{\text{iid}}{\sim} \mathcal{N}(\mathbf{0}, \frac{1}{5^2} \cdot \mathbf{I}_d)$. For any given number of features $P^\phi$, these are stacked to give a $n \times P^\phi$ dimensional random design matrix $\mathbf{\Phi}$, which is then used to solve the regression problem $\mathbf{y} = \mathbf{\Phi}\boldsymbol{\beta}$ by least squares. Following the methodology of [BHMM19] we perform one-vs-all regression for each class resulting in 10 regression problems. For principal component regression we standardize the design matrix, apply PCA, add a bias term, and finally perform ordinary least squares regression.

### E.2 "Peak-moving" experiments

In this section, we show that the first peak in generalization error can be arbitrarily (re)moved by changing *when*, i.e. at which value of $P^{leaf}$, $P^{boost}$ or $P^{PC}$, we transition between the two mechanisms for increasing the raw number of parameters in each of the three methods. We do so for two reasons: First, as can be seen in Fig. 11, it allows us to highlight that the switch between the two parameter-increasing mechanisms is indeed precisely the cause of the second descent in generalization error – in all three methods considered. Second, this also implies that the second descent itself is *not inherently caused by interpolation*: to the contrary, comparing the generalization curves in Fig. 11 with their train-error trajectories in Fig. 12, it becomes clear that such a second descent can also occur in models that have not yet and will not reach zero training error. Thus, the interpolation threshold does not itself cause the second descent – rather, it often coincides with a switch between parameter increasing mechanisms because parameters on the $P^{leaf}$, $P^{boost}$ and $P^{PC}$ axis can inherently not be further increased once interpolation is achieved.

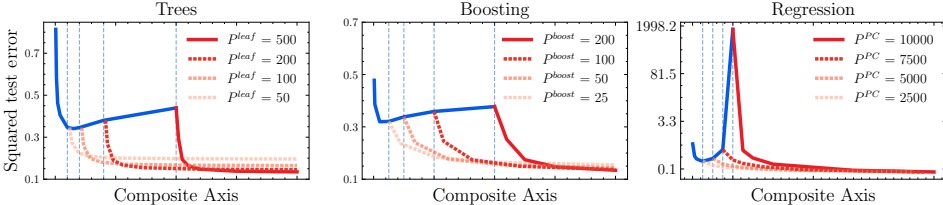

Figure 11: **Shifting the location of the double descent peak in generalization error by changing *when* we switch between the two parameter-increasing mechanisms.** Transitioning from $P^{leaf}$ to $P^{ens}$ at different values of $P^{leaf}$ in the tree experiments (left), transitioning from $P^{boost}$ to $P^{ens}$ at different values of $P^{boost}$ in the boosting experiments (middle) and transitioning from $P^{PC}$ to $P^{ex}$ at different values of $P^{PC}$ in the regression experiments (right).

Finally, we note that one could also create a generalization curve with arbitrarily many peaks and arbitrary peak locations (including peaks at raw parameter counts $p > n$) by switching between parameter-increasing mechanisms *more than once*. In Fig. 13, we show this for the linear regression example, where we switch between increasing parameters through $P^{PC}$ and $P^{ex}$ multiple times. This experiment is inspired by [CMBK21], who show one can "design your own generalization curve"

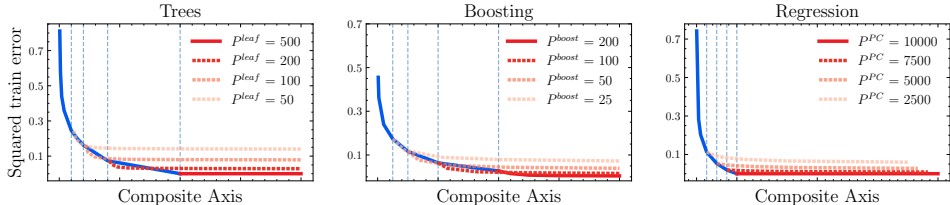

Figure 12: **Evolution of the training error for different transitions between the two parameter-increasing mechanisms.** Transitioning from $P^{leaf}$ to $P^{ens}$ at different values of $P^{leaf}$ in the tree experiments (left), transitioning from $P^{boost}$ to $P^{ens}$ at different values of $P^{boost}$ in the boosting experiments (middle) and transitioning from $P^{PC}$ to $P^{ex}$ at different values of $P^{PC}$ in the regression experiments (right).

in linear regression by controlling the data-generating process through the order by which newly (un)informative features are revealed. In our case, rather than inducing the change by changing the data, we manipulate the location of the peak by simply changing the sequence in which parameter counts are increased.

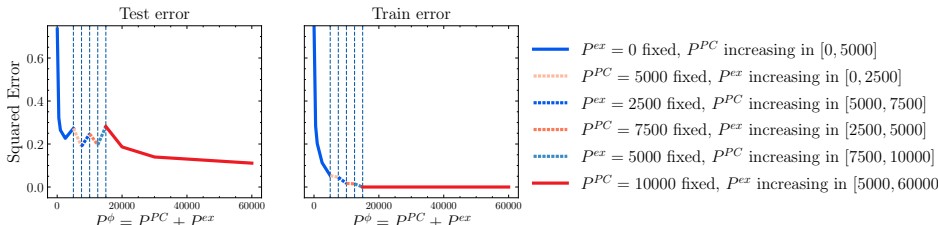

Figure 13: **"Design your own generalization curve".** We show using the regression example that a generalization curve with arbitrarily many peaks and arbitrary peak locations (including peaks at $P^\phi > n$) can be created simply by switching between increasing parameters through $P^{PC}$ and $P^{ex}$ multiple times (left). Train error remains monotonically decreasing (right).

### E.3  Plots of binary test error

In Figs. 14 to 16, we revisit Figs. 2, 4 and 5 of the main text by plotting 0-1 (or binary) loss achieved on the test-set – measuring whether the class with the largest predicted probability indeed is the true class – instead of summed squared loss across classes as in the main text. We observe qualitatively identical behavior in all experiments also for this binary loss.

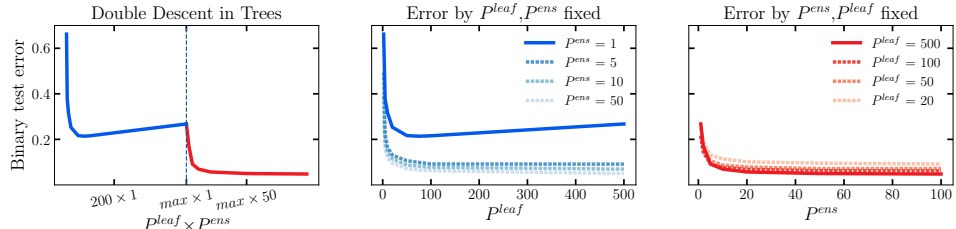

Figure 14: **Decomposing double descent in binary test error for trees.** Reproducing the tree experiments of [BHMM19] (left). Test error by $P^{leaf}$ for fixed $P^{ens}$ (center). Test error by $P^{ens}$ for fixed $P^{leaf}$ (right).

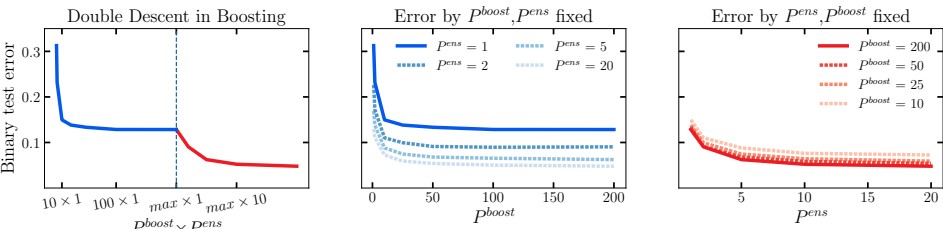

Figure 15: **Decomposing double descent in binary test error for boosting.** Reproducing the boosting experiments of [BHMM19] (left). Test error by $P^{boost}$ for fixed $P^{ens}$ (center). Test error by $P^{ens}$ for fixed $P^{boost}$ (right).

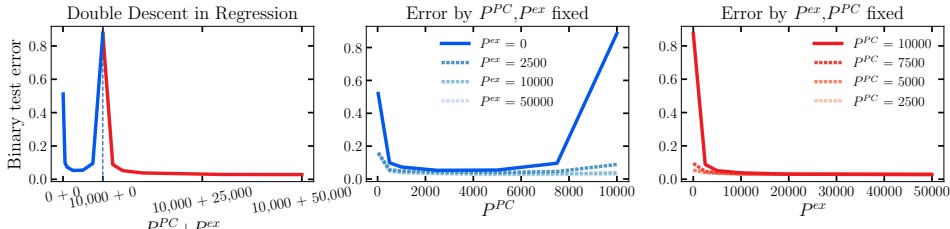

Figure 16: **Decomposing double descent in binary test error for RFF regression.** Reproducing the regression experiments of [BHMM19] (left). Test error by $P^{PC}$ for fixed $P^{ex}$ (center). Test error by $P^{ex}$ for fixed $P^{PC}$ (right).

### E.4 Plots of training loss

In Figs. 17 to 19 we provide plots of summed squared training losses associated with the plots of summed squared test loss in presented Figs. 2, 4 and 5 in the main text.

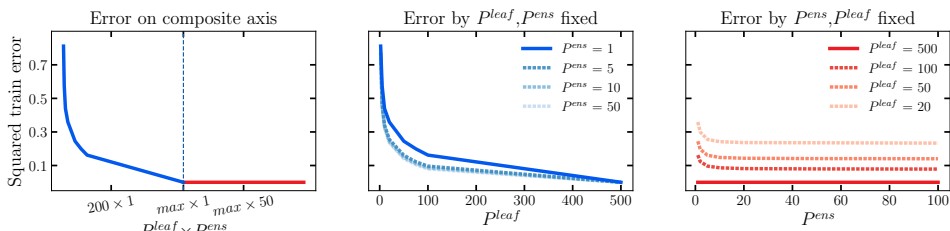

Figure 17: **Training error in the tree double descent experiments.** Training error against [BHMM19]'s composite axis (left). Train error by $P^{leaf}$ for fixed $P^{ens}$ (center). Train error by $P^{ens}$ for fixed $P^{leaf}$ (right).

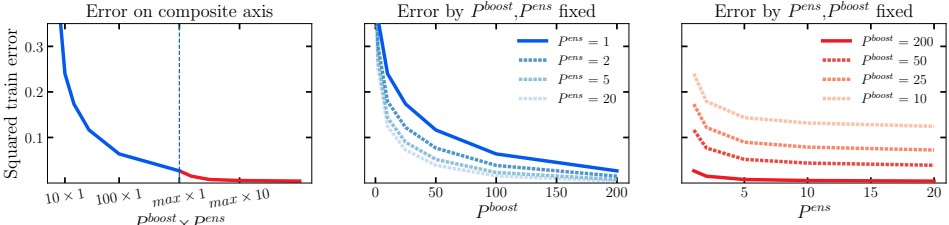

Figure 18: **Training error in the boosting double descent experiments.** Training error against [BHMM19]'s composite axis (left). Training error by $P^{boost}$ for fixed $P^{ens}$ (center). Training error by $P^{ens}$ for fixed $P^{boost}$ (right).

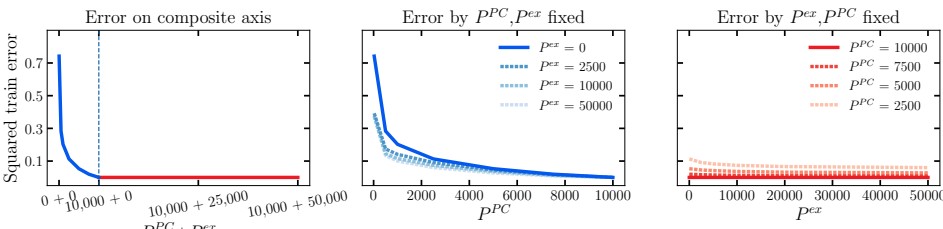

Figure 19: **Training error in the RFF regression double descent experiments.** Training error against [BHMM19]'s composite axis (left). Training error by $P^{PC}$ for fixed $P^{ex}$ (center). Training error by $P^{ex}$ for fixed $P^{PC}$ (right).

## E.5  Investigating applications of $p_{\hat{\mathbf{s}}}^{test}$ to model selection

In this section, we test whether a measure like $p_{\hat{\mathbf{s}}}^{test}$, measuring the effective parameters used on the test inputs, could be used for model selection purposes, possibly providing some route for redemption for parameter count based selection criteria like [Aka74, Mal73, Sch78] that explicitly trade off training error and parameters used (and importantly do not require access to held-out labels). We are motivated by empirical observations presented in Sec. 4. In particular, we note from Fig. 7 that across all three methods, there are models in the interpolation regime that perform very well in terms of generalization – and they happen to be those with the smallest $p_{\hat{\mathbf{s}}}^{\text{test}}$. This observation indicates that one straightforward model selection strategy when choosing between different hyperparameter settings that all achieve *zero training error* could be to compare models by their $p_{\hat{\mathbf{s}}}^{\text{test}}$.

Here, we test whether it is possible to exploit this in more general settings when there are multiple other hyperparameter axes to choose among. In particular, we consider tuning two other hyperparameters in gradient boosting, tree depth $P^{leaf} \in \{10, 50, 100, 200, 500, \max\}$ and learning rate $\eta \in \{0.02, 0.05, .1, .2, .3, .5, .85\}$. We do so in single boosted models (i.e. $P^{ens} = 1$), all trained until squared training error reaches approximately zero, i.e. we terminate after $P^{boost} \in [1, 500]$ rounds, where $P^{boost}$ is chosen as the first round (if any) where squared training error $l_{train}$ drops below $\epsilon = 10^{-4}$ (conversely, whenever $l_{train} > \epsilon$ for all $P^{boost} \leq 500$, we do not consider the associated ($P^{leaf}, \eta$) configurations an interpolating model).

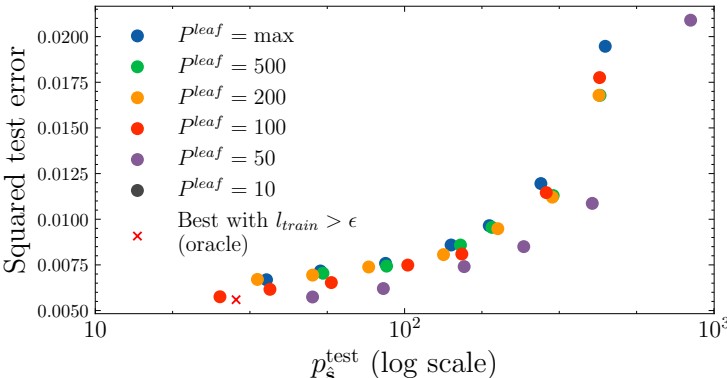

Figure 20: **Interpolating models with lower $p_{\hat{\mathbf{s}}}^{\text{test}}$ are associated with lower test error.** Test error vs $p_{\hat{\mathbf{s}}}^{\text{test}}$ for interpolating gradient-boosting models with of different ($P^{leaf}, \eta$)-combinations.

In Fig. 20 we observe that the intuition indeed carries over to this more general scenario and that interpolating models with lower $p_{\hat{\mathbf{s}}}^{\text{test}}$ are indeed associated with lower test error. In particular, choosing the model with lowest $p_{\hat{\mathbf{s}}}^{\text{test}}$ – which only requires access to test-time inputs $x$ but no label information $y$ – would indeed lead to the best test-performance among the interpolating models (with $l_{train} < \epsilon$). Further, using a red cross, we also plot the performance of the best non-interpolating model (with $l_{train} > \epsilon$) among the considered ($P^{leaf}, \eta$) configurations; this is selected among the models with less boosting rounds $P^{boost}$ by an oracle with access to test-time performance. We observe that this

non-interpolating oracle has essentially the same performance as the best interpolating model chosen by $p_{\hat{\mathbf{s}}}^{\text{test}}$.

## E.6 Results using other datasets

Below, we replicate our main experimental results using other datasets considered in [BHMM19], SVHN and CIFAR-10, exhibiting the same trends as the experiments on MNIST presented in the main paper.

### E.6.1 SVHN

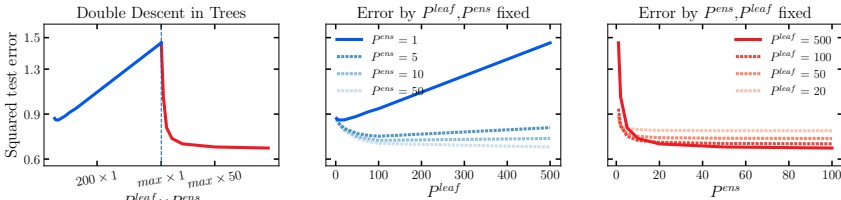

Figure 21: **Decomposing double descent for trees on the SVHN dataset.** Reproducing the tree experiment of [BHMM19] (left). Test error by $P^{leaf}$ for fixed $P^{ens}$ (center). Test error by $P^{ens}$ for fixed $P^{leaf}$ (right).

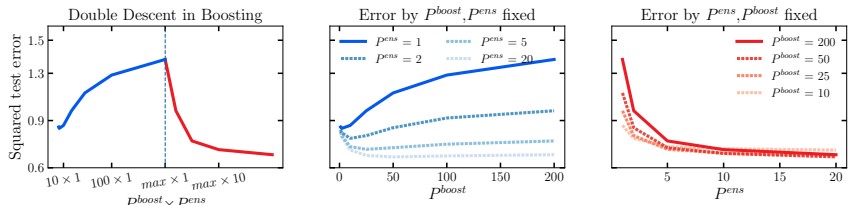

Figure 22: **Decomposing double descent for gradient boosting on the SVHN dataset.** Reproducing the boosting experiments of [BHMM19] (left). Test error by $P^{boost}$ for fixed $P^{ens}$ (center). Test error by $P^{ens}$ for fixed $P^{boost}$ (right).

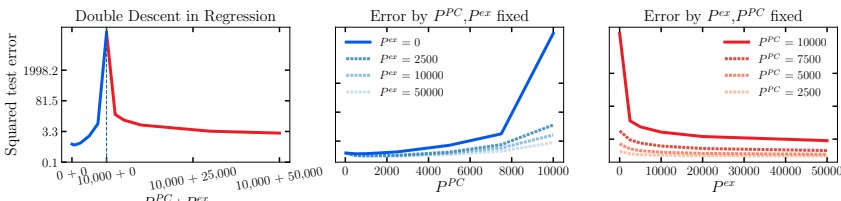

Figure 23: **Decomposing double descent for RFF Regression on the SVHN dataset.** Reproducing the RFF regression experiments of [BHMM19] (left). Test error by $P^{PC}$ for fixed $P^{ex}$ (center). Test error by $P^{ex}$ for fixed $P^{PC}$ (right).

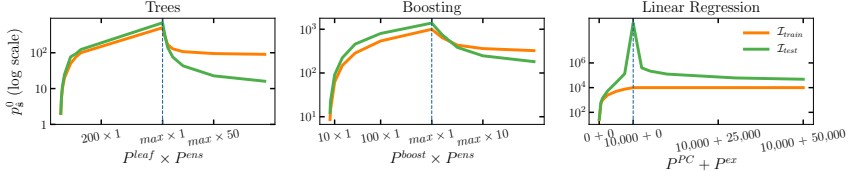

Figure 24: **The effective number of parameters does not increase past the transition threshold on the SVHN dataset.** Plotting $p_{\mathbf{s}}^{\text{train}}$ (orange) and $p_{\hat{\mathbf{s}}}^{\text{test}}$ (green) for the tree (left), boosting (center) and RFF-linear regression (right) experiments, using the original composite parameter axes of [BHMM19].

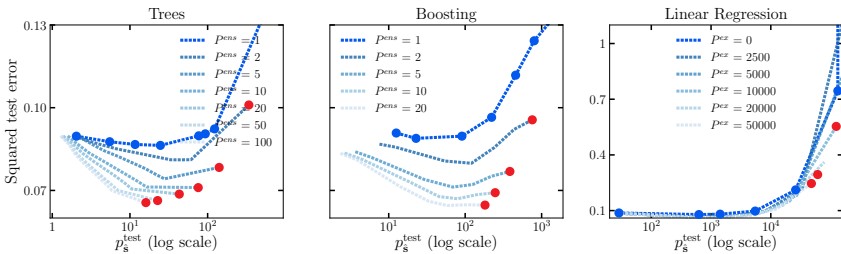

Figure 25: **Back to U (on the SVHN dataset).** Plotting test error against the *effective* number of parameters as measured by $p_{\mathbf{s}}^{\text{test}}$.

### E.6.2 CIFAR-10

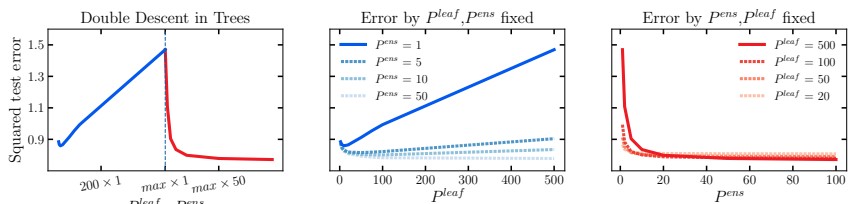

Figure 26: **Decomposing double descent for trees on the CIFAR-10 dataset.** Reproducing the tree experiment of [BHMM19] (left). Test error by $P^{leaf}$ for fixed $P^{ens}$ (center). Test error by $P^{ens}$ for fixed $P^{leaf}$ (right).

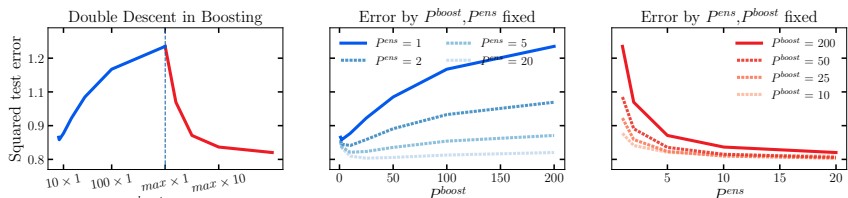

Figure 27: **Decomposing double descent for gradient boosting on the CIFAR-10 dataset.** Reproducing the boosting experiments of [BHMM19] (left). Test error by $P^{boost}$ for fixed $P^{ens}$ (center). Test error by $P^{ens}$ for fixed $P^{boost}$ (right).

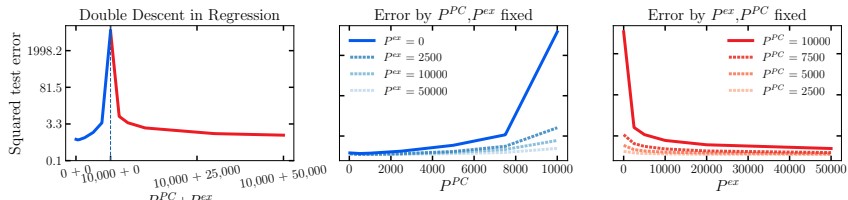

Figure 28: **Decomposing double descent for RFF Regression on the CIFAR-10 dataset.** Reproducing the RFF regression experiments of [BHMM19] (left). Test error by $P^{PC}$ for fixed $P^{ex}$ (center). Test error by $P^{ex}$ for fixed $P^{PC}$ (right).

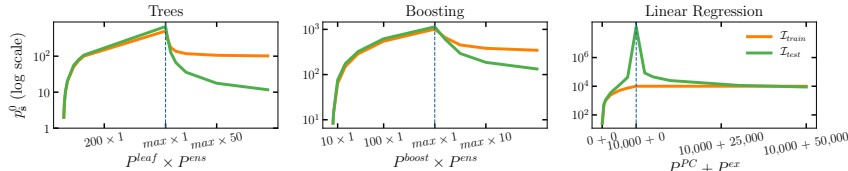

Figure 29: **The effective number of parameters does not increase past the transition threshold on the CIFAR-10 dataset.** Plotting $p_{\mathbf{s}}^{\text{train}}$ (orange) and $p_{\mathbf{s}}^{\text{test}}$ (green) for the tree (left), boosting (center) and RFF-linear regression (right) experiments, using the original composite parameter axes of [BHMM19].

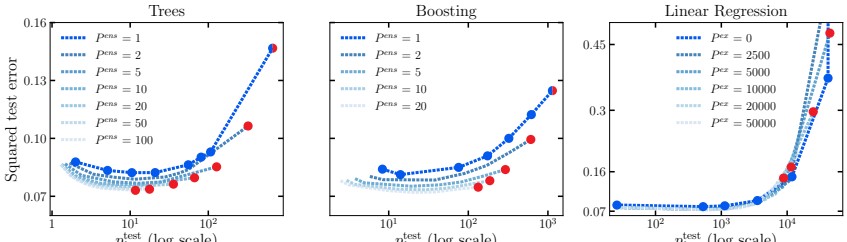

Figure 30: **Back to U (on the CIFAR-10 dataset).** Plotting test error against the *effective* number of parameters as measured by $p_{\mathbf{s}}^{\text{test}}$.

