# OpenReview forum: "A U-turn on Double Descent: Rethinking Parameter Counting in Statistical Learning"
_NeurIPS.cc/2023/Conference — NeurIPS 2023 oral_

### Official Review · Reviewer_tnvy · 2023-06-27

**Soundness:** 3 good
**Presentation:** 4 excellent
**Contribution:** 4 excellent
**Rating:** 7
**Confidence:** 4

**Summary:**

The paper aims to explain the well-known phenomenon of double-descent in the scope of several traditional ML methods (random forests, boosting and linear regression on random Fourier features). The main message of the paper (in the mentioned scope) is that what appears to be a double-descent on the first sight (using plots that count raw number of model parameters), might actually be a consequence of an intristincly two-dimensional parameter development composed of two traditionally behaved, complementary components. In all the studied cases, authors disentangle the parameter count into two parts - one accounting for a complexity of "a single model within the considered class", while the second accounting for something, that one could call "extent of statistical power in terms of some proxy of mixture of experts, or ensembles". The authors show that once the parameters are counted in this two-dimensional manner, the double-descent curve folds into two curves, of which one usually exhibits traditional U-shape and the other one exhibits an L-shape. For linear regression, authors consider a non-trivial disentanglement - into the number of basis vectors that are directly used for regression and the number of exceeding features.

In an independent part, authors consider models that fall into the category of smoothers (kNN, trees and forests, LR) and show that if one counts parameters in a more sensible way (in the paper via a specific, smoothers-tailored effective number of parameters measure), one can not even register a double-descent behavior anymore. The authors show that the test performance can be partially explained via the effective number of parameters (though it seems the number of ensembles/extra features should be taken into account too).

**Strengths:**

S1: The idea of decomposing the parameter count into two independent dimensions which unfold (and therefore partially explain) the well-known double-descent is (I believe) novel, surprising, important (at least within the scope of the studied models), interesting, neat and probably impactful. The analysis done for the models considered is strong, empirical evidence convincing and theoretical analysis for RFF-LR makes sense. The experiments in this section are well-chosen and well-done and to the point.

S2: As a part of the paper's exposition, another very important observation is made (around lines 192-195) - the double descent, being a consequence of different mechanisms (explained in the paper), is not strictly tight to the point where the model capacity enters the interpolation regime. We can obtain similar double-descent curves with breaking points far below the interpolation threshold.

S3: The quality of writing and the overall clarity and structure of the paper are well-above the average.



**Weaknesses:**

The weaknesses are roughly sorted by severity, starting with the most severe.

W1: The whole part 2 of the paper - explaining the double descent with alternative parameter counting - is not novel, as acknowledged by the authors. The idea of the double-descent appearing to happen only if raw parameters are displayed, but disappearing, or being more explainable, if effective number of parameters measure is used instead, is already done in several works cited by the paper. The authors defend the originality and novelty of this part and of the used effective parameter count as a tool for explaining double descent, within the scope of smoothers, by claiming that (i) unlike some of the previous work, it covers model classes that use trees and (ii) the proposed distinction between train and test time parameter count is essential to explaining the double-descent and (iii) the proposed parameter count does not require test-time labels, unlike the previous methods. However: (i) while this is true, the disadvantage is that this method is only usable for smoothers, and therefore it is very unclear how this can be used for, for instance, neural networks (unlike the measures used in the cited literature), which limits not only the scope of this part, but also, more importantly, the potential impact of this part. On the other hand, (ii) seems not to be well-reasoned. While it is true that for this particular measure and for smoothers and in-sample double-descent this seems to be the case, other complexity measures, such as the one considered in [1], but also those that [1] compares itself against, seem perfectly capable of explaining double-descent (for most of the models excluding trees) without this train-test distinction. Finally, (iii) seems not to even be true, as the method considered in [1] and the ones cited by [1] do not require the held-out / test data at all.

W2: Related to W1, the authors do not explain at all why the effective parameter count drops so significantly if we increase the number of ensembles/exceeding features (i.e. the secondary parameter count dimension). Perhaps it is not in the scope of the paper, but I would find it a significant strengthening of the paper, if this was included. In fact, the authors don't even explain (except in the in-sample RFF-LR case with the first dimension of parameters already being equal $n$), why the effective parameter count in the test time doesn't increase. I would, moreover, want to see an alternative to Figure 7, but with $P^{PCA}$ being fixed and $P^{Ex}$ being the variable for different base levels of $P^P{PCA}$, because with this it is not even clear what the relation between the $P^{Ex}$ and the effective parameter count with fixed (and potentially small) $P^{PCA}$ is.

W3: Though the authors provide a heuristic explanation on why increasing $P^{Ex}$ should improve the model's performance (by providing better basis features and better-conditioned feature matrices), this does not really explain the improvement in model's performance. The connection between the quality of basis features in terms of the properties of the basis feature matrix and the test performance is not clear. Information-theoretically, all the information necessary to predict $y$ should already be encoded in $X$, thus representing $X$ with increasing number of random features, just because some matrix becomes better conditioned, does not seem to causally explain the improvement in test performance.

W4: Related to W3, the authors do not at all discuss the apparent non-symmetry between the first and the second dimension. How come the second dimension of parameter count enjoys L-curve, while the first dimension still displays U-curve? Can you provide an explanation that would extend beyond the particularities of Appendix B and wrap all the considered models in a simple (or at least general-enough) way?

W5 (minor): I don't agree with line 290: "First, we observe that these results perfectly match the intuition developed in Part 1" - In part 1, there is no intuition built on why the measures of effective number of parameters should never increase beyond the first dimension of parameter count.

[1] Maddox, Wesley J., Gregory Benton, and Andrew Gordon Wilson. "Rethinking parameter counting in deep models: Effective dimensionality revisited." arXiv preprint arXiv:2003.02139 (2020).

**Questions:**

Q1: Why is the difference between test and train effective parameter count in Figure 6 for RFF-LR at the interpolation threshold so huge?

Q2: Do you have an intuition on how to extend the results of part 1 to neural networks? What would be the candidates for the two dimensions? What about other models? Is there a way to make this more model-agnostic?

Q3: Can you explain any of the questions raised in the "weaknesses" section?

Besides, I have a few comments:

C1: The statement in lines 159-160 seems to be incorrect. I don't see how min-norm solutions can project $y$ into a row space.

Summary:

Despite several significant concerns I have raised in the "weaknesses" section, the paper brings some truly novel and important ideas (most of the part 1). Therefore, I recommend accepting the paper. However, I would be more happy, if the authors addressed the issues raised in the "weaknesses" section, either in the rebuttal, or, more conveniently for them, by withdrawing the paper and submitting the improved version to another venue, producing a very strong submission.

Note: After seeing the author's response and having most of my concerns resolved, I have decided to raise my score from the original 6 to new 7.

**Limitations:**

I didn't find any explicit discussion of limitations, despite being marked present by the authors in the paper checklist. Perhaps a few sentences here-and-there can be interpreted as a discussion of limitations.

---

> ### Author Rebuttal · Authors · 2023-08-09
>
> We thank the reviewer for the very interesting, in-depth and constructive review! We were delighted by the assessment that Part 1 of our paper was "novel, surprising, important , interesting, neat and probably impactful" and, limited by space constraints, hope to resolve more pressing doubts about Part 2 in our response below.
>
> **Novelty of part 2 (W1).** We certainly agree that – as discussed in the related work section – we are not the first to use alternative ways of measuring model complexity to study double descent (DD) in general. Nonetheless, the approach we take in our paper is very different from existing work, resulting in significant novelty also in this section: in particular a) we are able to quantify and understand the effective parameters used by _different classes of methods_ than existing work, while making new connections to the literature on smoothing, b) we are the first to study and discover differences in the used effective parameters at training and testing inputs, and c) are able to uncover the ‘stacked U-curve’ phenomenon in Figure 7 (which, to the best of our knowledge, has not previously appeared in any related work) only _because our decompositions from part 1 revealed the two-axis nature of the original experiments_.
>
> **Comparison to [MBW20] (W1).** First, allow us to emphasize that we strongly believe our work to be _complementary_ to  [MBW20]: while their approach cannot natively handle models trained without explicit loss function, our $p^0_s$ may indeed seem less _naturally_ applicable to study neural networks (*see also the top-level rebuttal for more on neural nets!) – thus, depending on the use-case of interest, either may be more appropriate to use.  Second, regarding (W1, ii) we would like to highlight that a need to look beyond train-time effective parameters to understand DD can also arise when using [MBW20]’s approach.  To see this, note that linear regression is the only method considered in both our and [MBW20]’s paper; however, while we consider _unregularized_ regression like BHMM19, [MBW20]’s metric is really only well-motivated when one uses a prior with hyperparameter $\alpha > 0$ (i.e. corresponding to inclusion of a ridge penalty in a frequentist setting). If we let $\alpha$ -> 0, their effective parameter measure $\sum_{i=1}^p \lambda_i/(\lambda_i+\alpha)$ approaches $\sum_{i=1}^p 1[\lambda_i>0] = rank(X’X) = min(n, p)$ which is exactly equal to our $p^{train}_s$ for linear regression and hence constant once $p>n$. For the unregularized linear regression case, their approach therefore cannot explain DD and a train-test distinction is also necessary here. Finally, regarding (W1, iii) we agree with the review – [MBW20]’s measure does not require access to test time labels and we did not intend to imply that it does. We would like to clarify that the statement in l. 377  refers to selection criteria based on held-out loss (e.g. cross-validation) as discussed in l. 374 (and not [MBW20]’s work discussed earlier in the paper).
>
> **Understanding why $p^{test}_s$ drops along axis 2 (W2).** For tree-based methods, this is easiest to see: Intuitively, $||s(x_0)||^2$ here measures the non-uniformity in weight given to each training target when issuing predictions. In a single tree grown to full depth, each test example will fall into a leaf with a single training example – which has the maximum number of effective parameters. However, when ensembling multiple such trees, for test inputs $x_0$ in underdetermined regions of the feature space, weight is likely to be spread more uniformly as different random splits place $x_0$ in leaves with different training examples across the different trees in an ensemble. While a single tree acts like a 1-NN regressor, an ensemble of multiple trees can thus act more like a k-NN regressor (with k>1) for unseen inputs. For the regression case, note that – for a basis B constructed through PCA, we have that $||s(x\_0)||^2=||b(x\_0)(B’B)^{-1}B’||^2=b(x\_0)(B’B)^{-1}b’(x\_0)=\sum^{P^{PCA}}\_{p=1} b^2\_p(x\_0)/\lambda\_p$ as B has orthogonal columns and for each column $B\_p’B\_p=\lambda_p$ (where $\lambda_p$ is the pth eigenvalue of  $\Phi’\Phi$) by construction. On the training data, we have that, as expected, $\sum^n\_{i=1}||s(x\_i)||^2=\sum^n\_{i=1}\sum^{P^{PCA}\}_{p=1} b^2\_p(x\_i)/\lambda\_p = P^{PCA}$ as $\lambda\_p$ is also the empirical variance of the PCs ($\lambda\_p=n^{-1} \sum^n\_{i=1} b^2\_p(x\_i)$). However, on the unseen testing inputs, it is possible that previously unobserved large values along the low-variance directions with very small $\lambda_p$ will lead to terms that blow up the $s(x_0)$ immensely (which is exactly what we observe in our experiments). Thus, $||s(x_0)||^2$ for new inputs likely shrinks as the variances $\lambda_p$ of the included PCs grow – which – as we see empirically in Appendix B3 – happens as we increase $P^{ex}$ for fixed $P^{PCA}$. As suggested, we also included a plot of $p^{test}_s$ by $P^{ex}$ and $P^{PCA}$ separately (see the general rebuttal).  While our primary interest lies in understanding the DD phenomenon, we agree that interesting insights about the methods themselves can be gained through study of $p^{test}_s$ -- and will include some of the above into the updated manuscript, thank you for the suggestion!
>
> **Why would adding more $P^{ex}$ improve model performance? (W3)** Please refer to the general rebuttal, where we answer this for multiple reviewers!
>
> **Non-symmetry between axes (W4).** Our results provide some anecdotal evidence that the first axis can be best understood as a bias-reduction axis – it controls how well the training data can be fit (increasing its parameters reduces underfitting – only when the first axis is set to its maximum can interpolation be achieved). The second axis appears to predominantly achieve variance-reduction _at test-time_: it reduces ||s(x_0)||, reducing the impact of label noise by smoothing over more training examples.

---

> > ### Comment · Reviewer_tnvy · 2023-08-11
> >
> > Thank you for your great and instructive answers. I am reassured about most of the points that I raised. I think the authors provided satisfactory explanations. It would be great if the authors could provide these discussions (especially on W2, W3, W4) in the main text once they are allowed to do so. I have decided to raise my score to 7 since after seeing the answers the paper seems to be a clear accept for me.

---

### Official Review · Reviewer_trNU · 2023-07-05

**Soundness:** 3 good
**Presentation:** 4 excellent
**Contribution:** 4 excellent
**Rating:** 9
**Confidence:** 4

**Summary:**

This paper critically looks at the double descent phenomena, and it is shown that the number of parameters on the x-axis does not represent the complexity well for double descent curves, and in essence, causes the double descent. If a more appropriate complexity measure is adopted, in this paper, the effective number of parameters, the phenomena disappears, and we observe a typical U-shape or L-shape, as expected from learning theory.

Note: I did not have time to study the appendix in detail.

**Strengths:**

This paper addresses and resolves some paradoxes in the community surrounding the double descent phenenoma. As such, I believe it will have a high impact and is relevant for many researchers. Furthermore, the introduced complexity measure can be highly relevant for model selection tasks.

The analysis is straightforward and complete. Especially nice is that it revisits the original experiments of Belkin exactly and finds the root-causes of the behavior.

The figures are clear and intuitive, especially the color-coding with blue and red curves is very helpful to our understanding.

The analysis of linear regression seems to agree with some very early analysis by
S. Raudys and R. Duin, “Expected classification error of the
Fisher linear classifier with pseudo-inverse covariance matrix,”
Pattern Recognit. Lett., vol. 19, no. 5-6, pp. 385–392, 1998
who noted that dimensionality reduction indeed plays the suggested role. A more recent work that covers this is:
J. H. Krijthe and M. Loog, “The peaking phenomenon in semisupervised learning,” in S+SSPR. Springer, 2016, pp. 299–309
which can be slightly easier to read. However, the presentation in the current paper is much more clear and also covers the boosting and random forest experiments, while these papers only cover linear regression.

More connections and background on the double descent phenomena can be found in:
Viering, T., & Loog, M. (2022). The shape of learning curves: a review. IEEE Transactions on Pattern Analysis and Machine Intelligence.
For example, double descent can either be visible in learning curves or feature curves, a connection that is often overlooked.

Note that, historically, the double descent behavior was studied inbetween 1990 and 2019 (as also apparent from above), but that this was in a subcommunity that was not popular.

As an aside, we have attempted a similar analysis as the one given in the paper via VC bounds and other similar measures for regression, but they generally are not fine grained enough for the analysis required. I am very happy the authors did manage to unmask the paradox via the proposed complexity measure.

**Weaknesses:**

none

**Questions:**

Are there any connections with the proposed metric of Generalized Effective Number of parameters and generalization? E.g. are there any upperbounds on the generalization that can be given in terms of the proposed measure? That would be an added strength

---

> ### Author Rebuttal · Authors · 2023-08-09
>
> We would like to thank the reviewer for the very interesting, in-depth and constructive review! We are delighted by the reviewer’s overwhelmingly positive review of our paper, in particular the assessment that “it will have a high impact and is relevant for many researchers”! We also put much effort into making the paper intuitive and easy to follow, and are especially excited to hear that this was appreciated. (Responding to the aside, we were also very interested in the reviewer’s own experience with applying other measures of complexity to this problem, and in this light especially thrilled by the review’s favourable assessment of our own approach!).
>
> Finally, we are grateful for the additional references included in the review –  we will include these in the updated version of  the extended literature review in Appendix A.
>
> With regards to the question on providing bounds for generalization using our effective parameter measure, we completely agree with the reviewer that this will be a very natural -- albeit challenging -- next step that will drive further, more crisp, understanding of the effect of effective parameters on generalization. It is not something we have investigated deeply so far, as we were focussed on finding an intuitive and empirical resolution to the double descent phenomenon, but we hope that it may be possible to borrow further insights from the vast literature on smoothing to make progress in this direction in the future!

---

### Official Review · Reviewer_4SoX · 2023-07-11

**Soundness:** 3 good
**Presentation:** 3 good
**Contribution:** 3 good
**Rating:** 7
**Confidence:** 2

**Summary:**

This paper proposes a new perspective for understanding the double descent behavior in non deep learning models: Once the number of raw parameters goes pass the interpolation threshold, the effective parameter number actually stops increasing and if one keeps increasing the number of parameters, the underlying model class will be modified. The authors analyze three examples from existing literature: Decision tree, gradient boosting and linear regression. In particular, the authors perform a novel analysis on overparameterized linear regression shows that the increment of Fourier feature dimensions would first increase the number of PCA regression dimensions, which follows the classic U curve structure, and then increase the number of excess feature dimension, which results in a L shape of test error. However, studying the effective dimension size requires model-specific derivation, therefore the author proposes a unified framework for studying the effective parameter size in different models using techniques from smoothers. With, the proposed measurement as the x-axis, the double descent shape on y-axis disappears, which aligns with the experiment results for individual models.

**Strengths:**

- The paper is well written and well motivated.

- The analysis and proposed measurement of effective number of parameter is technically sound.

- The relationship with existing literature is thoroughly discussed.

**Weaknesses:**

- If I understand correctly, the proposed measurement needs to be computed using test samples, which is not as straightforward as other metrics that only requires training sample, e.g. the measurement proposed in [MBW20].

**Questions:**

- The authors show that for decision tree, gradient boosting, and linear regression, there exists an axis in model complexity which follows "bigger is better". Does this axis exist for all types of models?

- How does the proposed measurement fit into unsupervised learning models where there does not exist `y` in the task.

**Limitations:**

See Weaknesses

---

> ### Author Rebuttal · Authors · 2023-08-09
>
> We thank the reviewer for the thoughtful review -- we are very delighted by the very positive assessment of our work! Below, we respond to the points raised in the review in turn.
>
> **The need to evaluate effective parameters on test samples.** We appreciate this reviewer raising this point as we believe this to be, in fact, a key strength of our work. Specifically, we would like to highlight that, as shown in Fig. 6, we _can_ compute our metric on the training examples too -- but observed this measurement of effective parameters without consideration of the test input distribution to be insufficient to explain generalization performance and were therefore motivated to address this limitation in our calculation of effective parameters. This becomes apparent when we compare to the measure in [MBW20] as suggested. Note that [MBW20]'s metric is motivated from a Bayesian setting where one uses a prior with parameter $\alpha>0$ (corresponding to ridge regularization in a frequentist setting), while we consider unregularized regression. If we let $\alpha$ approach 0, their effective parameter measure $\sum\_{i=1}^p \lambda\_i/(\lambda\_i+\alpha)$, where $\lambda\_i$ are the eigenvalues of X'X (computed using training examples), approaches $\sum\_{i=1}^p 1[\lambda\_i>0] = rank(X'X) = min(n, p)$ which is exactly equal to what we measure with $p^{train}_s$ for linear regression -- and hence constant once p>n. Therefore, like the case of applying our metric to the training data (as shown in Fig 6), this metric would fail to capture the decrease in complexity during the second descent. This highlights precisely the _strength_ of our metric being that it _can_ also be applied to test data (note: this is without requiring access to labels).  Only through this application to test data, as discussed on L262-281, we may better extend beyond the fixed design setting into the modern machine learning regime where we are primarily interested in generalization to _unseen_ covariates.
>
> **Do these two types of axes exist for all types of models?** While we intentionally focused here on the three specific methods under investigation, we would certainly consider it an interesting and fruitful direction for future research to understand whether other ML methods also have a similar distinction into parameter axes along which generalisation performance evolves in particular patterns! Our results provide anecdotal evidence that the first parameter axes may be responsible for bias reduction – it controls how well the training data can be fit –, while the second parameter axes drive variance reduction for test-time predictions; studying how other methods fit into this intuition would certainly be an interesting next step!
>
> **How does the proposed measurement fit into unsupervised learning?** Although model complexity is typically primarily considered in the supervised setting, notions of model complexity are certainly valuable in the unsupervised setting too. Since the smoother matrix upon which we measure effective parameters is a transformation from ground truth labels to predictions one might similarly consider the unsupervised setting in which we have a transformation from input space into some (usually compressed) representation. A simple example might be the family of principal component analysis methods. Another interesting case for consideration might be semi-supervised learning in which we have access to some unlabelled data which clearly aligns with our measure. While further exploration of these settings is clearly outside the scope of this paper, considering how one might transfer the results from this and other similar works beyond the supervised setting would certainly make for an interesting research direction.

---

> > ### Comment · Reviewer_4SoX · 2023-08-13
> >
> > Thanks for the detailed response! I remain recommending acceptance for this paper.

---

### Official Review · Reviewer_Uot3 · 2023-07-12

**Soundness:** 3 good
**Presentation:** 4 excellent
**Contribution:** 3 good
**Rating:** 7
**Confidence:** 3

**Summary:**

This paper studies the double descent phenomenon where it has been suggested that optimal generalization performance is achieved as model complexity gets extremely large and training data is perfectly overfit. However, this work suggests that true model complexity is not described simply by the number of parameters. This claim is backed by replications of experiments from a foundational double descent paper, where the main result is empirical evidence and a theoretical explanation on how linear regression in high dimension reduces to a lower dimension problem on principle component features. Lastly, the authors propose a way to measure the number of effective parameters for each replicated experiment.

**Strengths:**

- The paper is well-written and the motivation for the work is well-defined
- Proposition 1 is an interesting and novel observation and its proof is rigorous from what I can tell
- The area of overparameterized machine learning is of very high relevance today

**Weaknesses:**

An obvious weakness of the paper is that it does not use modern neural network models. I recognize that work on linear models is necessary as a foundation and the authors do discuss this limitation in the discussion, however it is troubling that there is no proper justification/commentary on the connection between the work and modern architectures.

**Questions:**

I'm curious how well your linear regression explanation extends to other statistical models like in BLLT20 and HMRT22.

**Limitations:**

The authors adequately address the limitations.

---

> ### Author Rebuttal · Authors · 2023-08-09
>
> We thank the reviewer for the thoughtful review -- we are delighted by the positive assessment of our presentation, relevance and our insights into the linear regression case study!
>
> **Connections to Neural Networks.** We completely agree that – as discussed in l. 356ff. – understanding double descent in neural networks is a natural and very interesting next step – which we consider highly nontrivial, and to hence fall beyond the scope of this paper. In addition to the possible connections already stated in our conclusion, we believe that one very interesting and promising starting point for extending our work to the neural network setting is to use our approaches for understanding double descent in linear models and apply them to neural networks trained  in the _lazy regime_ (see e.g. ‘On Lazy Training in Differentiable Programming’, Chizat et al, NeurIPS19), which essentially act like models that are linear in features defined through the gradients of the model at initialisation. Nonetheless, allow us to emphasize that we strongly believe that understanding double descent in simpler models (e.g. smoothers), as we do in this paper,  is of value to the community as is – both in its own right _and_ to provide a foundation for follow up work to understand this phenomenon in more complex models – exemplified also by the numerous papers published recently studying double descent in linear models exclusively.
>
> **How does our linear regression explanation extend to BLLT20 and HMRT22?** Where BLLT20 and HMRT22 also study min-norm linear regressions, our basic arguments naturally apply: Also in their studies, due to the use of the min-norm solution,  there is an implicit change in the parameter-increasing mechanism at p=n and thus once p>n, the number of truly determined directions remains constant at n and $p^{train}_s=n$ by construction. While $p^{train}_s$ is easy to determine as it is independent of the problem characteristics and simply equal to min(n,p),  $p^{test}_s$ depends on the data characteristics thus studying theoretically how this would evolve would likely involve different assumptions on the data-generating mechanism. Studying the interaction between our work and specific data generating process would definitely be an interesting avenue for future work!

---

> > ### Comment · Reviewer_Uot3 · 2023-08-10
> >
> > I thank the authors for their response. I have decided to raise my score to a 7.

---

### Official Review · Reviewer_vbD3 · 2023-07-13

**Soundness:** 3 good
**Presentation:** 3 good
**Contribution:** 3 good
**Rating:** 7
**Confidence:** 3

**Summary:**

This paper studies the double descent phenomenon focusing on the complexity axis used to display the phenomenon. For trees and boosting, they show that double descent is a result of peculiar axes used by one previous work and that an alternative formulation of complexity removes the phenomenon. They then propose a generalized measure for the effective number of parameters for both trees, boosting, and lienar models which recovers a more classical U-shape.

**Strengths:**

- The paper is clearly written
- The reparameterization of complexity for trees and boosting is clear and needed + for linear models is well-motivated
- The introduced generalized effective number of parameters based on smoothers is interesting and seems to work well

**Weaknesses:**

- The evidence for double descent in trees is very limited and weak (one main figure and one appendix figure in BHMM19); the authors may want to dedicate more time to the rest of the paper rather than refuting this phenomenon
- The experimental settings shown are very limited, would be nice to see less synthetic settings, particularly in Fig 7 to see how well the proposed complexity measure holds


**Questions:**

- A deeper comparison with related work would be helpful, particularly MBW20, DLM20, and DSYW20.
- The authors should connect their analysis in Section 3 based on principal components  with the more popular analyses using ridge regression rather than limiting PCs (e.g. that in HMRT22). The empirical observations are extremely similar and the connection is fairly close, as ridge regression can (often) be seen as a smooth eversion of PCA regression.


**Limitations:**

- See above

---

> ### Author Rebuttal · Authors · 2023-08-09
>
> We thank the reviewer for the thoughtful review -- we are delighted by the positive assessment of our presentation, complexity reparametrization and effective parameter measure! We hope to resolve doubts about any weaknesses in our point-by-point response below.
>
> **Inclusion of boosting and trees.** We are delighted by the deep interest in our results on the linear regression case study and effective parameters, and certainly agree that they might be of most individual interest to the community. Nonetheless, allow us to attempt to convince you why we consider the case studies on trees and boosting an important part of our paper. The goal of our work was to study the appearance of double descent outside of deep learning generally, motivated by [BHMM19] who suggest they “provide evidence for the existence and _ubiquity of double descent for a wide spectrum of models_” (abstract). The experiments that we replicate were said to “give empirical evidence that the families of functions explored by boosting with decision trees and random forests also show similar generalization behavior to that of neural nets” (p.4) – this constitutes the only example of non-deep double descent except for linear regression that we are aware of, which is why we consider these results important to include. We certainly agree that the experimental setup is a lot less natural than the regression case study – in fact, this is precisely what inspired our paper in the first place: our investigation of the tree-based experiments (where the two-axes decomposition may be intuitively apparent to some readers) is what motivated our search for orthogonal parameter axes in the linear regression case (leading us to distinguish between PC and excess features), and we thus consider it an important pre-requisite for our deeper analyses.
>
> **Experimental settings.** The goal of our work was to reanalyse and provide explanations for the existing evidence for non-deep double descent -- of which BHMM19 (and the datasets within) provided the broadest account. Therefore, following closely the experimental setup in their paper was a key design choice for our work, as this focus allowed us to really zoom-in and provide in-depth analyses of the original experiments and findings.However, we are happy to provide results on an additional non-image, real-world dataset of different modality for this reviewer (also, note that replication of our complete analyses using some additional datasets were already included in our App E.6). To do so we selected MiniBooNE, a dataset from the particle physics community in the Fermilab in which 50 features represent a stream of muon neutrinos that are fired and recorded by a detector which measures the presence of electron neutrinos (signal) among the muon neutrinos (noise). The goal is to classify observations as signal or background noise. This task has been recently recognised as a recommended benchmark task in researching machine learning methods for unstructured data [1].
>
> We process the data just as in the other experiments and repeat the analysis for the RFF-regression. The results are included in the attached PDF in the top level rebuttal where we observe the same double descent effect along the raw axes, which, when using our proposed measure of effective parameters, again reduces into a familiar U-shaped complexity curve consistent with our previous experiments.
>
> [1] Grinsztajn, Léo, Edouard Oyallon, and Gaël Varoquaux. "Why do tree-based models still outperform deep learning on typical tabular data?." Advances in Neural Information Processing Systems 35 (2022): 507-520
>
>
> **Connections to the ridge regression results of HMRT22** The results in our work are entirely consistent with and complimentary to those of [HMRT22]. This is particularly highlighted in their Fig. 1 where they find the usual double descent shape in (ridgeless) min-norm regression but _no double descent_ in optimally tuned ridge regression (i.e. where the regularization strength is selected to optimize validation performance). Interestingly, at $\gamma = 1$, where the number of features is equal to the number of examples, standard analysis based on just parameter counts would suggest performance should be poor. This is because, as shown in our work, this is typically where _effective parameters_ are highest. However, this is counteracted by optimally tuned ridge regression which would automatically select for higher regularization at this point and therefore reduces the implied effective parameters – thus removing the double descent behaviour entirely. This is an excellent illustration of the importance of measuring _effective parameters_ over raw parameters as advocated for in our work. We therefore thank the reviewer for suggesting that our section on extended related work in appendix A could be expanded even further; we will include additional discussions of the connections between ridge regression and PCR as well as a discussion of work on regularization in the context of the double descent phenomenon!
>
> **Further discussion of MBW20, DLM20, and DSYW20.**  We will include an additional discussion of the effective parameter measures used by DLM20, and DSYW20 into Appendix D, where we already discuss MBW20’s measure in more detail.

---

> > ### Comment · Reviewer_vbD3 · 2023-08-11
> > **Reviewer response**
> >
> > I thank the authors for their rebuttal and have decided to raise my score to 7. I still believe the authors should contextualize their claims more early on in the paper to note the known results for ridge regression.

---

### Official Review · Reviewer_WN4r · 2023-07-21

**Soundness:** 4 excellent
**Presentation:** 3 good
**Contribution:** 3 good
**Rating:** 7
**Confidence:** 4

**Summary:**

The paper analyzes the seeming appearance of double descent in non-deep models, and finds various ways in which it disappears when thinking more carefully about how the parameters are scaled. Trees and boosted models see double descent because parameters start increasing along a different and more effective dimension (ensemble size) once p=n. Linear regression sees double descent because the min-norm optimization solution used in prior work functions differently before and after p=n. And all of these methods no longer see double descent when the "effective number of parameters" is used as the x-axis instead of the raw number of parameters.

**Strengths:**

I greatly enjoyed reading the paper. The findings on linear regression are very interesting and contribute towards the understanding of the important phenomenon of double descent. It is well-presented, with concise and easy-to-understand figures and clear writing. It is likely to encourage follow-on work in the field.

**Weaknesses:**

The portions on trees/boosting and linear regression sit a bit unevenly together. In my view, the trees/boosting section doesn't actually tell us much about double descent since we're talking about independently trained ensembles, which is a different issue altogether. Meanwhile, the linear regression bit is very interesting.

The connections between min-norm solutions, SVD, and PCA (and the Moore-Pensrose pseudoinverse) are reasonably well understood (see e.g. these class notes http://www.sci.utah.edu/~gerig/CS6640-F2012/Materials/pseudoinverse-cis61009sl10.pdf), so I'd question the novelty of the theory a bit. However, the application of the theory to understand the problem was excellent.

The section on effective number of parameters didn't spend enough time analyzing the relationship between raw parameter count and effective parameter count (and any implications thereof), which seems worth dwelling a bit from the purpose of understanding double descent (more so than showing the lack of double descent using that metric as the x-axis).

I think the paper would be improved with more connections to related work and related ideas, e.g. regularization and particularly ridge regression (due to its connections with PCA), work on multi-layer linear networks and PCA (e.g. Baldi and Hornik 1988, "Neural Networks and Principal Component Analysis: Learning from Examples Without Local Minima), connections between neural network optimization and min-norm solutions (which you do give a brief mention to, but seems quite relevant)

**Questions:**

I like the paper and applaud your work, but as a take-it-or-leave-it suggestion I would consider whether a reframing around your core contributions (insights about overparameterized linear regression) vs. being a rebuttal to another paper (forces simple and less interesting critiques about ensembles to take up space) would help. There's interesting stuff in the appendix (e.g. condition number analyses of bases) that could certainly fit in the main paper if you end up going that route.

One major question on my mind after reading this paper is what it "means" to find a richer basis among the input features and why that helps. Do you have any insights on this? Are there applications to preconditioning? How closely tied is it to the choice of RFF features, vs. alternate setups where e.g. there are p>n features by default?

Per the 'weakness' above, can you add the relationship/analysis of raw parameter count and effective parameter count?

---

> ### Author Rebuttal · Authors · 2023-08-09
>
> We thank the reviewer for the in-depth and constructive review! We were delighted by the assessment that our paper is "interesting", "well-presented " and "likely to encourage follow-on work in the field", and, limited by space constraints, hope to resolve doubts about any weaknesses in our response below!
>
> **Inclusion of boosting and trees.** We are delighted by the deep interest in our results on the linear regression case study! We agree that these may be of most individual interest to the community and will use the additional space in the updated manuscript to prioritise moving some of the additional linear regression results from the appendix to the main text. Nonetheless, allow us to attempt to convince you why we consider the case studies on trees and boosting an important part of our paper. The goal of our work was to study the appearance of double descent outside of deep learning generally, motivated by [BHMM19] who suggest they “provide evidence for the existence and _ubiquity of double descent for a wide spectrum of models_” (abstract). The experiments that we replicate were said to “give empirical evidence that the families of functions explored by boosting with decision trees and random forests also show similar generalization behavior to that of neural nets” (p.4) – this constitutes the only example of non-deep double descent except for linear regression that we are aware of, which is why we consider these results important to include. We certainly agree that the experimental setup is a lot less natural than the regression case study – in fact, this is precisely what inspired our paper in the first place: our investigation of the tree-based experiments (where the two-axes decomposition may be intuitively apparent to some readers) is what motivated our search for orthogonal parameter axes in the linear regression case (leading us to distinguish between PC- and excess features), and we thus also consider this part an important pre-requisite for our deeper analyses.
>
> **Connections between min-norm solutions, SVD, and PCA.** We did not intend to imply that we are the first to notice a connection between these concepts; rather, we indeed consider our application of these insights to the double descent context our main novelty. However, to the best of our knowledge, the specific result derived in Prop 1 has not previously been formalized in this way in the literature and provided an essential tool for our analysis in Sec. 3. We have added a line to our discussion of these topics further emphasising this point.
>
> **Additional analyses of raw and effective parameter count.** We thank the reviewer for the suggestion to include further analyses of the raw parameter axes and the effective parameter count, as we agree that this provides interesting new insights into the behaviour of the ML methods. In particular, per the reviewer's request, we created figures plotting the effective parameters against the two separate parameter axes (see Figures attached to general rebuttal), which allows to make even more clear the role of the two parameter axes: along the first axes, effective parameters monotonically increase while along the second axes, effective parameters monotonically decrease. Thus, this further confirms that (as we had stated in l. 317) the second axis acts by ‘decreasing the effective complexity implied by each value’ along the first axis).  We agree that these results provide evidence that compliments Figs 6 & 7  – and will include some of the insights into the updated manuscript once the additional content page becomes available!
>
> **Understanding basis quality implied by $P^{ex}$.** Please refer to the general rebuttal, where we answer this for multiple reviewers!
>
> **Additional connections to related work.** We thank the reviewer for suggesting that our section on extended related work in appendix A could be expanded even further; we will include additional discussions of the connections between ridge regression and PCR as well as a discussion of work on regularization and the double descent phenomenon! We will also expand upon the connections between neural networks and our work.

---

> > ### Comment · Reviewer_WN4r · 2023-08-16
> >
> > Thank you for the detailed response! After reviewing all of your comments, especially those relating to insights about the richer bases and about the effective parameter count, as well as promises to add more contextualization and lit review, I am willing to bump up my score to a 7.

---

### Author Rebuttal · Authors · 2023-08-09

We would like to once more thank all reviewers for their time and effort put into the review process -- we thoroughly enjoyed reading the interesting and constructive reviews we received. We were delighted by the overwhelmingly positive response to our paper, and are thrilled that all 6 reviewers agreed that our study of the double descent phenomenon will be of interest to the NeurIPS community! We also received numerous stimulating questions (extending far beyond what we could hope to answer within the scope of a single conference paper) and hope that future work will follow up on many of interesting directions highlighted in the reviews!

Below, we provide some responses to questions raised by multiple reviewers, and have also attached additional figures that -- as requested -- display the relationship between the two raw axes and effective parameter axes across the different methods (Figs 1 and 2) and experiments using an additional dataset of different modality (Fig 3).

**How could we extend this work to neural networks?** (Reviewers  Uot3, tnvy) In addition to the possible connections already stated in our conclusion  l. 356ff., we believe that one very interesting and promising starting point for extending our work to the neural network setting would be to use our approaches for understanding double descent in smoothing models  and apply them to neural networks trained  in the _lazy regime_ (see e.g. ‘On Lazy Training in Differentiable Programming’, Chizat et al, NeurIPS19), as these networks essentially act like models that are linear in features defined through the gradients of the model at initialisation -- to these linearised networks some of our arguments from the linear regression case study may extend very naturally.

**What is the relationship between the two raw parameter axes and effective parameter count?**  (Reviewers WN4r, tnvy) In the attached Figs. 1&2, we find that along the respective first axes ($P^{leaf}, P^{boost}, P^{PCA}$) effective parameters monotonically increase while along the second axes ($P^{ens}, P^{ex}$), effective parameters monotonically decrease. Thus, this further confirms that (as we had stated in l. 317) the second axis acts by ‘decreasing the effective complexity implied by each value’ along the first axis). We agree that these results provide interesting evidence that compliments Figs 6 & 7 – and will include some of the insights into the updated manuscript once the additional content page becomes available!

**How exactly does adding more $P^{ex}$ improve model performance?** (Reviewers WN4r, tnvy) We agree that this is a very interesting (and nontrivial!) question, whose answer may extend beyond the scope of our original investigation. Below, we discuss our current understanding, but think that exploring this question further would constitute a fruitful direction for future work.

Reviewer tnvy raised this question while noting that "all the information necessary to predict y should already be encoded in X" and that thus representing it with an increasing number of features (regardless of the conditioning of some matrix) would not seem to explain the improvement in performance. To intuitively address this point, we note that, for example, once p>n, adding additional features indeed does not add any new information for predicting the observed (training inputs) y. Yet, in such underspecified settings, different representations that make the same predictions on training inputs can lead to wildly different generalisation performance to new inputs– despite being indistinguishable on the training data. A basis construction step thus essentially serves to choose between models that all explain y equally well (and hence cannot be distinguished based on supervised loss), akin to an unsupervised pre-processing step. The _implicit inductive bias_ encoded in the step we uncover essentially consists of constructing and choosing the top-$P^{PCA}$ features that _capture the directions of maximum variation_ in the data.

Within this inductive bias, the role of $P^{ex}$ appears to be that -- as more excess features are added -- the variation $\lambda_p$ captured by each of the top-$P^{PCA}$ principal components is likely to increase (which we indeed see empirically in Appendix B.3). This increase in the $\lambda_p$ in turn is likely to _decrease_ the variance of the regression predictions on unseen inputs because $Var(\hat{y}) = Var(s(x\_0)y)=\sigma^2\sum^{P^{PCA}}\_{p=1} b^2\_p(x\_0)/\lambda\_p$ (where $b(x_0)=(b_1(x_0), ..., b_{P^{PCA}}(x_0))$ is the representation of the input point using the PCs) can explode on the unseen testing inputs when previously unseen large values $b^2\_p(x\_0)$ occur along the low-variance directions with very small $\lambda\_p$ at test time (which is exactly what we observe in our experiments).

Still, using the directions of maximum variation is certainly not guaranteed to be optimal in all applications (Jolliffe, "A note on the use of principal components in regression.",  1982) but it tends to be an effective inductive bias in practice as noted in Tukey (“ Exploratory data analysis.”, 1977) who suggested that the high variance components are likely to be more important for prediction unless nature is "downright mean". However, note that we do not intend to make a claim that increasing $P^{ex}$ is always effective – rather, we are simply uncovering that this appears to be the underlying effect in the RFF experiments.

---

### Decision · Program_Chairs · 2023-09-21

**Decision:**

Accept (oral)

**Comment:**

The paper provides valuable insights into the paradox of double descent curve. Observations on the double descent curve postulate there is a gap between theoretical uniform convergence rate and practical observations. While multiple studies on this observation, the current paper introduces an innovative perspective aimed at bridging the disparity between theory and practice through the introduction of a more precise understanding of complexity. The paper shows that the phenomenon of double descent arises from the amalgamation of two distinct effects within a single plot, potentially leading to confusion. This outcome underscores the need for future practical and theoretical investigations, with the potential to ultimately resolve the paradox inherent in the double descent curve.